# Decoupling Classifier for Boosting Few-shot Object Detection and Instance Segmentation

**Bin-Bin Gao**[1]    **Xiaochen Chen**[1]    **Zhongyi Huang**[1]    **Congchong Nie**[1]    **Jun Liu**[1]

**Jinxiang Lai**[1]    **Guannan Jiang**[2]    **Xi Wang**[2]    **Chengjie Wang**[1]

[1]Tencent YouTu Lab      [2]CATL

## Abstract

This paper focus on few-shot object detection (FSOD) and instance segmentation (FSIS), which requires a model to quickly adapt to novel classes with a few labeled instances. The existing methods severely suffer from bias classification because of the missing label issue which naturally exists in an instance-level few-shot scenario and is first formally proposed by us. Our analysis suggests that the standard classification head of most FSOD or FSIS models needs to be decoupled to mitigate the bias classification. Therefore, we propose an embarrassingly simple but effective method that decouples the standard classifier into two heads. Then, these two individual heads are capable of independently addressing clear positive samples and noisy negative samples which are caused by the missing label. In this way, the model can effectively learn novel classes while mitigating the effects of noisy negative samples. Without bells and whistles, our model without any additional computation cost and parameters consistently outperforms its baseline and state-of-the-art by a large margin on PASCAL VOC and MS-COCO benchmarks for FSOD and FSIS tasks.[1]

## 1 Introduction

Fully supervised deep convolutional neural network have achieved remarkable progress on various computer vision tasks, such as image classification [15], object detection [12, 3, 29], semantic segmentation [22, 2] and instance segmentation [14] in recent years. However, the superior performance heavily depends on a large amount of annotated images. In contrast, humans can quickly learn novel concepts from a few training examples. To this end, a few-shot learning paradigm [8] is presented, and its goal aims to adapt novel classes when only providing a few labeled examples (instances). Unfortunately, existing few-shot models are still far behind humans, especially for few-shot object detection (FSOD) and few-shot instance segmentation (FSIS).

Various methods have been proposed to tackle the problem of the FSOD and FSIS. The earlier works [34, 39] mainly follow meta-learning paradigm [9] to acquire task-level knowledge on base classes and generalize better to novel classes. However, these methods usually suffer from a complicated training process (episodic training) and data organization (support query pair). The recent transfer-learning (fine-tuning) methods [33, 35, 31, 1, 28] significantly outperforms the earlier meta-learning ones. Furthermore, it is more simple and efficient. These transfer-learning methods mainly follow a fully supervised object detection or instance segmentation framework, e.g., Faster-RCNN [30] or Mask-RCNN [14]. Therefore, it may be suboptimal for few-shot scenario.

The PASCAL VOC [4] and MS-COCO [21] are widely used to evaluate the performance of object detection or instance segmentation. Under a fully supervised setting, the model can be well-trained

---

Corresponding author: B.-B. Gao (csgaobb@gmail.com) and C. Wang (jasoncjwang@tencent.com).

[1]https://csgaobb.github.io/Projects/DCFS.

36th Conference on Neural Information Processing Systems (NeurIPS 2022).

on these two datasets because all interest objects are almost completely labeled. Under an instance-level few-shot scenario, however, we find that there is a large number of instances that are missing annotations as shown in Fig. 1 (detailed discussion in Sec. 3.2). The reason is that the community considers an instance as a shot for controlling the number of labeled instances when building instance-level benchmarks. This is different from image-level few-shot image classification [32] because there are generally multiple instances in an image for instance-level few-shot learning. In fact, missing (partial or incomplete) label learning is more difficult and challenging, especially instance-level few-shot scenarios. It requires that learning algorithms deal with training images each associated with multiple instances, among which only partial instances are labeled; while the common supervised learning typically assumes that all interest instances are fully labeled. In some real-world applications, such as open-vocabulary object detection [40], it is almost impossible to label all instances, and thus there still may exist some instances left to be missing labeled. In addition, it is more friendly and convenient for users to label partial instances than all ones even in few-shot settings.

Most methods [42, 25] have been developed to address missing label (partial label or incomplete label) learning but mainly focus on image-level multi-class or multi-label classification. To address the instance-level missing label issue, some recent works have attempted to regard the missing (unlabeled) instances as hard negative samples and re-weight [37] or re-calibrate [41] their losses. However, these works still only focus on general object detection. For instance-level few-shot recognition, it may result in biased classification and thus limit the generalization ability of novel classes using the model trained on these mislabeled datasets if we don't take any action.

Recently, one work closely related to ours is the state-of-the-art DeFRCN [28] which decouples Faster-RCNN to alleviate the foreground-background confusion between base pre-training and novel fine-tuning in FSOD. It also can be interpreted from a missing label perspective. Here, we could view fine-tuning few-shot learning paradigm as a domain adaption procedure from base to novel. In this procedure, a few-shot detector may suffer from foreground-background confusion because one background proposal (negative class may be potential novel class) in the base learning stage will become foreground (positive class) in the novel fine-tuning phase. To mitigate the label conflict between the two domains, DeFRCN decouples RCNN and RPN by stopping gradient backpropagation of RPN in Faster-RCNN. Different from the missing label of cross-domain in DeFRCN, we focus on the missing label issue only in the novel (or balanced base-novel) fine-tuning stage. Another recent work, Pseudo-Labelling [26, 18] mines the missing labeled instances for increasing the number of positive training samples and reducing the biased classification. Unfortunately, this method may lead to a chicken-and-egg problem–we need a good few-shot detector to generate good pseudo labels, but we need good few-shot annotations to train a good few-shot detector. Unlike this work, our method completely avoids using any pseudo-label information.

In this paper, we propose a simple decoupling method to mitigate the biased classification issue. Specifically, we firstly decouple the standard classifier into two parallel heads, positive and negative ones. Then, these two heads independently process clear positive and noisy negative samples with different strategies. Our contributions are summarized as follows:

- We rethink FSOD and FSIS from the perspective of label completeness and discover that existing transfer-learning few-shot methods severely suffer from bias classification because the missing label issue naturally exists in instance-level few-shot scenarios. To be best of our knowledge, this is the first to propose missing label issue in FSOD and FSIS.
- To mitigate the bias classification, we propose a simple but effective method that decouples the standard classifier into two parallel heads to independently process clear positive samples and noisy negative ones. Without bells and whistles, the proposed decoupling classifier can be taken as an alternative to the standard classifier in state-of-the-art FSOD or FSIS models.
- Comprehensive experimental results on PASCAL VOC and MS-COCO show that our approach without any additional parameters and computation cost outperforms state-of-the-art both on FSOD and FSIS tasks.

## 2 Related Work

**FSOD** aims to recognize novel objects and localize them with bounding boxes when only providing a few training instances on each novel class. Existing works can be roughly grouped into two families, meta-learning and transfer-learning, according to training paradigm. The meta-learning

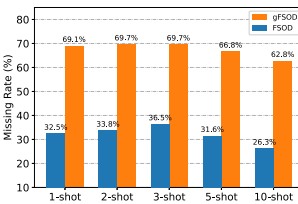 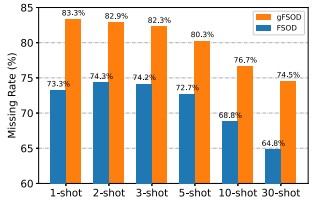 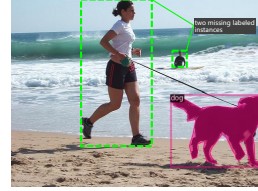

| (a) PASCAL VOC | (b) MS-COCO | (c) a one-shot labeled image |

Figure 1: The proportion of missing instances in the training set for FSOD and gFSOD on (a) PASCAL-VOC and (b) MS-COCO datasets. It can be observed that there is a high missing rate in each shot, especially for the gFSOD. In (c), two "person" instances present in the one-shot labeled image, but they are mislabeled.

methods [34, 17, 39, 5, 38, 19, 20, 43, 16, 13] use episodic training to acquire task-level knowledge on base classes and generalize better to novel classes. The transfer-learning methods [33, 35, 31, 43, 1, 28] generally utilize two-stage training strategy first pre-training on base classes and then fine-tuning on novel classes, which have significantly outperformed many earlier meta-learning approaches. Recent FSCE [31] shows that the degradation of detection performance mainly comes from misclassifying novel instances as confusable classes, and they propose a contrastive proposal encoding loss to ease the issue. Similar to the FSCE, FADI [1] explicitly associates each novel class with a semantically similar base class to learn a compact intra-class distribution. DeFRCN [28] decouples Faster-RCNN [30] to alleviate the foreground-background confusion between pre-training and fine-tuning stage.

**FSIS** needs to not only recognize novel objects and their location but also perform pixel-level semantic segmentation for each detected instance. Many FSIS approaches typically use the FSOD framework, e.g., Mask-RCNN [14], it is built on the Faster-RCNN by adding a mask segmentation head. Therefore, most FSIS methods generally follow the FSOD learning paradigm, i.e., meta-learning [23, 39, 7, 24] or transfer-learning [10]. Siamese Mask-RCNN [23] and Meta-RCNN [39] commonly compute embeddings of support images and combine them with those features of query images produced by a backbone network. Their difference is the combination strategy, e.g., subtraction in [23] and concatenation in [39]. These works only focus on the performance of novel classes and ignore that of base classes. In real-world applications, we expect that one few-shot model not only recognizes novel classes but also remembers base classes. The recent iMTFA [10] introduces incremental learning into FSIS and propose incremental FSIS task.

## 3   Methodology

### 3.1   Few-shot Object Detection and Instance Segmentation Setting

Given an image dataset $D = \{X_i, Y_i\}_{i=1}^N$, where $X_i$ denotes the $i$-th image and $Y_i$ is its corresponding annotation. For object detection, $Y_i = \{b_k, c_k\}_{k=1}^M$, where $b_k$ and $c_k$ represent bounding box coordinates and category of the $k$-th instance presented image $X_i$, respectively. For instance segmentation, $Y_i$ includes pixel-level mask $m_k$ annotation beyond category and bounding box ones, i.e., $Y_i = \{b_k, m_k, c_k\}_{k=1}^M$. Under few-shot learning setting, these annotations can be grouped into two sets, base and novel classes, denoted as $C_{base}$ and $C_{novel}$. Note that the base and novel classes are non-overlapping (i.e., $C_{base} \cap C_{novel} = \emptyset$).

FSOD or FSIS aims to detect/segment novel class instances through training a model based on plenty of labeled instances on a set of base classes and a few instances (e.g, 1, 2, 3 and 5) on each novel class. Note, FSOD or FSIS only focuses on recognizing novel class instances but ignores base class ones. It is impractical from the perspective of many real-world applications because people always expect that a few-shot model is capable of not only recognizing novel classes but also remembering base classes. To this end, generalized FSOD and FSIS (abbreviated as gFSOD and gFSIS) [11, 27, 6, 10] stresses that a good few-shot learning system should adapt to new tasks (novel classes) rapidly while maintaining the performance on previous knowledge (base classes) without forgetting.

As mentioned above, transfer-learning FSOD and FSIS methods mainly consist of two stages, i.e., pre-training on base classes and fine-tuning on novel classes. The former is trained on plenty of labeled instances in $C_{base}$ while the latter only uses a few labeled instances in $C_{novel}$. For generalized

few-shot learning, the main difference is novel stage training which uses few labeled instances in both $C_{base}$ and $C_{novel}$. Note that the labeled instances are abundant at the base stage, but there are only a few labeled instances at novel stage both FSOD/FSIS and gFSOD/gFSIS. Therefore, most instances are unlabeled (missing labeled) as only $K$ instances are provided at the fine-tuning stage. This is, in fact, reasonable from the perspective of data privacy (incremental learning).

## 3.2   Revisiting Object Detection and Instance Segmentation

We focus on transfer-learning FSOD and FSIS methods because they are more simple and effective compared to meta-learning ones. As we know, Faster-RCNN and Mask-RCNN are very popular and powerful solutions as two-stage stacking architecture for fully supervised object detection and instance segmentation. We firstly describe the two-stage detector and segmenter. In general, the first stage is designed to generate class-agnostic region proposals which can be formulated as:

$$\mathcal{F}_{s1}(\theta_{s1}; \cdot) = \mathcal{F}_{ROI} \circ \mathcal{F}_{RPN} \circ \mathcal{F}_{EF}(\cdot), \tag{1}$$

where the $\theta_{s1}$ is all the parameters of the first stage. Specifically, an input image $X_i$ is firstly fed into a backbone network to Extract high-level Features ($\mathcal{F}_{EF}$). Then, a Region Proposal Network (RPN) is adopted to generate candidate regions based on these extracted features ($\mathcal{F}_{RPN}$). Finally, all these region proposals are pooled into fixed size feature maps using a Region of Interest (ROI) pooling module ($\mathcal{F}_{ROI}$) for the following stage. The structure of the second stage varies depending on the specific task. For object detection, the ROI features of *sampled* region proposals will be parallelly fed into two heads for performing box classification and regression, that is

$$\mathcal{L}_{s2}^{OD} = \mathcal{L}_{CLS}(\theta_{cls}; \cdot) + \mathcal{L}_{REG}(\theta_{reg}; \cdot), \tag{2}$$

where $\theta_{cls}$ and $\theta_{reg}$ are the parameters of classification and regression head, respectively. Instance segmentation method (e.g., Mask-RCNN) follows the second stage structure of object detection and applies it to three heads, i.e., box classification, box regression and mask segmentation, that is

$$\mathcal{L}_{s2}^{IS}(\cdot) = \mathcal{L}_{CLS}(\theta_{cls}; \cdot) + \mathcal{L}_{REG}(\theta_{reg}; \cdot) + \mathcal{L}_{SEG}(\theta_{seg}; \cdot), \tag{3}$$

where $\theta_{seg}$ is parameters of the mask segmentation head. Given an input image $X_i$ and its corresponding annotation $Y_i = \{b_k, c_k\}_{k=1}^{M}$ or $Y_i = \{b_k, m_k, c_k\}_{k=1}^{M}$, Eqs. 2 and 3 can be jointly optimized end-to-end by minimizing $\mathcal{L}_{s2}^{OD}(\cdot)$ and $\mathcal{L}_{s2}^{IS}(\cdot)$, which follows a multi-task learning paradigm. For simplicity, we omit the RPN learning in Eqs. 2 and 3.

**Missing Label Issue.** We can obtain a powerful model by minimizing Eqs. 2 and 3 when all interesting instances are completely labeled in a large-scale image dataset $D$. The completeness of annotation at label space is, in fact, a general precondition for most fully supervised learning algorithms. However, it is not satisfactory for a few-shot learning scenario. The reason is that only a few instances are manually labeled in given training images, and a lot of potential instances may be presented but unlabeled. From the perspective of practical application, it is natural and unavoidable to miss annotations when facing a few-shot setting. As shown in Fig. 1 (c), we can see that there are at least three instances including two "person" and one "dog" in this image, but only the "dog" instance carrying annotation (box, mask, category), and other two "person" instances are unlabeled (i.e., missing labels).

In order to quantitatively measure the proportion of missing labeled instances, we compute the average missing rate for each shot on PASCAL VOC and MS-COCO benchmark datasets as shown in Fig. 1. Firstly, it can be seen that there is a high missing rate on each shot. For example, 74.3% novel class instances potentially present on 3-shot MS-COCO training images, but they are not given any annotations. Secondly, the missing rate is further increased under a generalized few-shot setting. For example, the missing rate increases nearly two times of few-shot setting on the PASCAL VOC. Despite some efforts try mining these missing labeled instances in a semi-supervised manner for boosting few-shot performance, the fully supervised loss with Eqs. 2 and 3 may result in a suboptimal solution when some instances are unlabeled. To be best of our knowledge, this is the first to propose the missing label issue in few-shot object detection and instance segmentation.

**Biased Classification Issue.** The Eqs. 2 and 3 can be optimized using sampled ROI features and their labels as mentioned above. The sampling operation performs box matching between "ROI features" and "annotation", and assigns training labels (positive or negative) to the corresponding ROIs. Here, the positive (foreground) ROIs are sampled from object proposals that have an IoU overlap with the

ground-truth bounding box at a threshold (e.g., 0.5), while negative (background) ROIs are sampled from the remaining proposals. The classification head is trained based on these sampled positive and negative ROIs. Different from the classification head, the box regression, and mask segmentation head learning are only associated with positive ROIs.

The annotations of positive ROIs are accurate, e.g., the annotations of the "dog" instances in Fig. 1 (c). However, the sampled negative ROIs may be noisy because of the missing label issue under the few-shot setting. For example, those two "people" instances will be assigned to negative labels (i.e., background) if they are sampled in Fig. 1 (c), according to the above label assignment strategy. This will make the standard classification head confused with positive and noisy negative samples. On one hand, the model is correctly guided to recognize positive objects because all positive samples are accurate. On the other hand, the model may be misguided by noisy negative samples and thus incorrectly recognize positive objects as background. Therefore, the bias classification may happen when meeting the missing label issue, especially in a few-shot scenario. Furthermore, it potentially limits the generalization ability to adapt to the novel class quickly and efficiently.

### 3.3 Decoupling Classifier to Mitigate the Bias Classification

**Standard Classifier.** We assume that $x \in \mathbb{R}^{C+1}$ is the predicted logit of a sampled ROI feature obtained from Eq. 1 and its corresponding class label vector is $y \in \mathbb{R}^{C+1}$, where there are $C$ foreground categories and one background class, and $y_i$ is 1 if the corresponding proposal belongs to the $i$-th category, 0 otherwise. Then, we use a softmax function to transform it into a probability distribution, that is

$$\hat{p}_i = \frac{\exp(x_i)}{\sum_t \exp(x_t)}. \tag{4}$$

The cross-entropy loss is used as the measurement of the similarity between the ground-truth $y$ and predicted distribution $\hat{p}$, that is

$$\mathcal{L}_{\text{CLS}} = -\sum_{i=0}^{C} y_i \log(\hat{p}_i). \tag{5}$$

Note that the standard classifier (Eqs. 4 and 5) may confuse with clear positive and noisy negative samples in few-shot scenario.

**Decoupling Classifier.** In order to process positive and negative samples differentially, we decouple the standard classifier into two heads, i.e., positive (foreground) head and negative (background) head, which are formulated as

$$\mathcal{L}_{\text{CLS}} = \mathcal{L}_{\text{CLS}}^{\text{fg}} + \mathcal{L}_{\text{CLS}}^{\text{bg}}. \tag{6}$$

Here, the positive and negative heads are responsible for positive and negative samples, respectively. Considering that the labels of positive samples (foreground) are accurate, we can use cross-entropy loss (i.e., Eq. 5) for all positive instances. The labels of those negative examples may be noisy because of the missing label issue. Therefore, it is not reasonable to employ normal cross-entropy loss for training the negative head. Note that these negative examples are generally sampled from those object proposals that have a maximum IoU overlap with the ground truth bounding box at an interval $[0, 0.5)$, and thus we can infer that they may not belong to the ground truth class, although we don't know their true category. We expect that the bias classification would be mitigated if the negative head performs learning only between few-shot labeled categories and the background class. To this end, we first obtain an image-level multi-label with instance-level few-shot annotation of a training image, and denote it as $m = [m^0, m^1, \cdots, m^{C-1}, m^C]^T$, where $m^i$ is a binary indicator, and $m^i$ is 1 if the image is labeled with the $i$-th category, 0 otherwise. Note that $m^C = 1$ indicates that each image at least contains a background class. Then, we can obtain a constrained logit $\bar{x}$ conditioned on the $m$, that is

$$\bar{x}_i = m_i x_i. \tag{7}$$

Substituting Eq. 7 into the softmax function Eq. 4 yields:

$$\bar{p}_i = \frac{\exp(m_i x_i)}{\sum_t \exp(m_t x_t)}. \tag{8}$$

We compute cross-entropy loss between $\bar{p}$ and the corresponding ground truth $y^{\text{bg}}$, that is

$$\mathcal{L}_{\text{CLS}}^{\text{bg}} = -\sum_{i=0}^{C} y_i^{\text{bg}} \log(\bar{p}_i), \tag{9}$$

where $y_C^{\mathrm{bg}}=1$ and $y_{i\neq C}^{\mathrm{bg}}=0$.

**Optimization with Decoupling Classifier.** Considering the joint optimization goal (Eq. 2 and 3) of object detection and instance segmentation, the optimal parameters $\Theta$ is determined by minimizing Eq. 2 and 3, where $\Theta = \{\theta_{s1}, \theta_{cls}, \theta_{reg}\}$ in object detection, and $\Theta = \{\theta_{s1}, \theta_{cls}, \theta_{reg}, \theta_{seg}\}$ in instance segmentation. For simplicity, we only consider the optimization for the classification head and omit the box regression and mask segmentation head in the following analysis. $\theta_{cls}$ is updated by a gradient descent step, that is

$$\theta_{cls} \leftarrow \theta_{cls} - \lambda \frac{\partial \mathcal{L}_{\mathrm{CLS}}}{\partial \theta_{cls}}, \tag{10}$$

where $\lambda$ is the learning rate. Note that we have decoupled the standard classifier into positive and negative learning in Eq. 6. We firstly analyze the $\theta_{cls}$ optimization for positive head. According to the chain rule in Eq. 4 and 5, we have

$$\frac{\partial \mathcal{L}_{\mathrm{CLS}}^{\mathrm{fg}}}{\partial \boldsymbol{x}} = \hat{\boldsymbol{p}} - \boldsymbol{y}^{\mathrm{fg}}. \tag{11}$$

Then, the derivative of $L_{\mathrm{CLS}}^{\mathrm{fg}}$ with respect to $\theta_{cls}$ is

$$\frac{\partial \mathcal{L}_{\mathrm{CLS}}^{\mathrm{fg}}}{\partial \theta_{cls}} = (\hat{\boldsymbol{p}} - \boldsymbol{y}^{\mathrm{fg}}) \frac{\partial \boldsymbol{x}}{\theta_{cls}}. \tag{12}$$

Combining the negative head in Eq. 9 and Eqs. 8 and 7, we can similarly obtain derivative of $L_{\mathrm{CLS}}^{\mathrm{bg}}$ with respect to $\theta_{cls}$ is

$$\frac{\partial \mathcal{L}_{\mathrm{CLS}}^{\mathrm{bg}}}{\partial \theta_{cls}} = \boldsymbol{m}(\bar{\boldsymbol{p}} - \boldsymbol{y}^{\mathrm{bg}}) \frac{\partial \boldsymbol{x}}{\theta_{cls}}, \tag{13}$$

where $\boldsymbol{y}^{\mathrm{bg}}$ is the ground truth label vector of a negative sample. Comparing Eqs. 12 with 13, we can see that the parameters $\theta_{cls}$ of the classification head will be updated with different ways for positive and negative examples. For the positive head, the gradient is updated in each dimension of the class space. But for the negative head, the gradient is limited in some special dimensions because of the introduced $\boldsymbol{m}$ and thus the bias classification may be alleviated.

In order to further understand Eqs. 12 and 13, we give a visualization example for decoupling classifier as illustrated in Fig. 2. Note that we use Gaussian normal distribution for prediction and ground truth distribution for intuition. Here, we take the one-shot labeled image in Fig. 1 (c) for example, where only the "dog" instance is labeled. We assume that one "person" instance is sampled as negative sample, and it will be mistaken as background class (ground truth). Due to the proposed decoupling classifier, the optimization of the "person" instance is constrained between the "dog" and "background" as shown in Fig. 2 (b) and doesn't affect the predictions of other categories such as the "person" class.

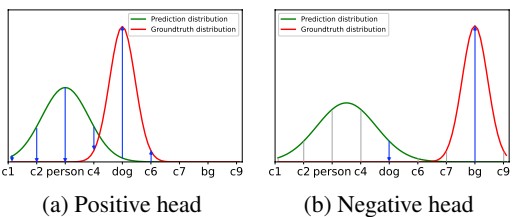

(a) Positive head      (b) Negative head

Figure 2: Illustration of the gradient of decoupling classifier, where the blue arrow represents the gradient direction. (a) illustrates the gradient propagation on the positive head, and (b) reveals that the gradient propagation is constrained between few-shot labeled class (e.g., dog) and the background and thus the bias classification is mitigated. Best viewed in color and zoom in.

## 4 Experiments

In this section, we empirically evaluate the proposed method for FSOD/gFSOD and FSIS/gFSIS tasks and demonstrate its effectiveness by comparison with state-of-the-art methods.

### 4.1 Experimental Setup

**Datasets.** We follow the previous works and evaluate our method on PASCAL VOC [4] and MS-COCO [21] datasets. For a fair comparison, we use the same data splits given in [33, 28].

**PASCAL VOC** covers 20 categories, which are randomly split into 15 base classes and 5 novel classes. There are three such splits in total. In each class, there are $K$ (1, 2, 3, 5, 10) objects for

Table 1: FSIS performance for Novel classes on MS-COCO. The superscript $\dagger$ indicates that the results are our re-implementation. The results are averaged over all 10 seeds and the best ones are in bold, the same below.

| Methods | | Tasks | 1 | | 2 | | 3 | | 5 | | 10 | | 30 | |
|---|---|---|---|---|---|---|---|---|---|---|---|---|---|---|
| | | | AP | AP50 | AP | AP50 | AP | AP50 | AP | AP50 | AP | AP50 | AP | AP50 |
| Meta R-CNN [39] | ICCV 19 | | - | - | - | - | - | - | 3.5 | 9.9 | 5.6 | 14.2 | - | - |
| MTFA [10] | CVPR 21 | | 2.47 | 4.85 | - | - | - | - | 6.61 | 12.32 | 8.52 | 15.53 | - | - |
| iMTFA [10] | CVPR 21 | Det | 3.28 | 6.01 | - | - | - | - | 6.22 | 11.28 | 7.14 | 12.91 | - | - |
| Mask-DeFRCN† [28] | ICCV 21 | | 7.54 | 14.46 | 11.01 | 20.20 | 13.07 | 23.28 | 15.39 | 27.29 | 18.72 | 32.80 | 22.63 | 38.95 |
| **Ours** | | | **8.09** | **15.85** | **11.90** | **22.39** | **14.04** | **25.74** | **16.39** | **29.96** | **19.33** | **34.78** | **22.73** | **40.24** |
| Meta R-CNN [39] | ICCV 19 | | - | - | - | - | - | - | 2.8 | 6.9 | 4.4 | 10.6 | - | - |
| MTFA [10] | CVPR 21 | | 2.66 | 4.56 | - | - | - | - | 6.62 | 11.58 | 8.39 | 14.64 | - | - |
| iMTFA [10] | CVPR 21 | Seg | 2.83 | 4.75 | - | - | - | - | 5.24 | 8.73 | 5.94 | 9.96 | - | - |
| Mask-DeFRCN† [28] | ICCV 21 | | 6.69 | 13.24 | 9.51 | 18.58 | 11.01 | 21.27 | 12.66 | 24.58 | 15.39 | 29.71 | 18.28 | 35.20 |
| **Ours** | | | **7.18** | **14.33** | **10.31** | **20.43** | **11.85** | **23.24** | **13.48** | **26.67** | **15.85** | **31.33** | **18.34** | **35.99** |

few-shot training. And the PASCAL VOC 2007 testing set is used for evaluation. The dataset is only used to evaluate FSOD task. We report the Average Precision (IoU=0.5) for novel classes (AP50).

**MS-COCO** contains 80 classes. The 20 categories presented in the PASCAL VOC are used as novel classes and the remaining 60 categories are used as base classes. We train a few-shot model based on $K$ (1, 2, 3, 5, 10, 30) instances for each class and evaluate on the MS-COCO validation set. This dataset has been widely used to evaluate the performance of FSOD and FSIS. We report Average Precision (IoU=0.5:0.95), Average Precision (IoU=0.5) on novel classes for FSOD and FSIS settings. In addition, we also report AP and AP50 for overall classes, base classes, and novel classes under gFSOD and gFSIS settings, respectively.

**Experimental Details.** The experiments are conducted with Detectron2 [36] on NVIDIA GPU V100 on CUDA 11.0. We use standard Faster-RCNN with ResNet-101 backbone extracted features from the final convolutional layer of the 4-th stage for few-shot object detection, which is the same as DeFRCN [28]. For instance segmentation, we add a mask prediction head at the ROI of the Faster-RCNN. For model training, we employ a two-stage transfer-learning approach: first training the network on the base classes with pre-trained by ImageNet and then fine-tuning on $K$-shots for every class. The SGD is used to optimize our network end-to-end with a mini-batch size of 16, momentum 0.9, and weight decay $5e^{-5}$ on 8 GPUs. The learning rate is set to 0.02 during base training and 0.01 during few-shot fine-tuning. Following the previous work [33], all experimental results are averaged over 10 seeds. For a fair comparison with DeFRCN [28], we also report the average results over 10 times repeated runs on `seed0`.

**Strong Baseline.** For FSOD and gFSOD, we take the state-of-the-art DeFRCN [28] as a strong baseline of our method. For FSIS and gFSIS, we extend DeFRCN similarly to how Mask-RCNN extends Faster-RCNN, i.e., adding a mask prediction head at the ROI of the DeFRCN and keeping others as same as DeFRCN, and thus we call it Mask-DeFRCN. Our method only replaces the standard classifier in DeFRCN and Mask-DeFRCN with our decoupling classifier. Therefore, DeFRCN and Mask-DeFRCN can be taken as our strong baseline for FSOD/gFSOD and FSIS/gFSIS tasks.

## 4.2 Comparison with the State-of-the-Art

**Few-shot Instance Segmentation on the MS-COCO.** *Our method (simple decoupling classifier) outperforms the state-of-the-art on the MS-COCO in both FSIS and gFSIS settings.* The main results on MS-COCO are reported in Table 1 and 2 for FSIS and gFSIS, respectively. Based on the experiment results, we have the following observations: **1)** The strong baseline (i.e., Mask-DeFRCN) outperforms all the state-of-the-art methods for FSIS and gFSIS; **2)** Our method consistently outperforms the baseline Mask-DeFRCN for FSIS; Compared to FSIS, our method has significant improvements for gFSIS. This is not surprised because the missing rate of gFSIS always is higher than that of FSIS as shown in Fig. 1 (b). This indicates our method is capable of addressing the missing label issue under few-shot setting; **3)** Our method has a better advantage, especially in low-shot (1-, 2-, 3-shot), and thus it is very suitable for a few-shot scenario.

On one hand, the missing rate of low-shot is generally higher than that of high-shot so that it leaves more improvement space for our method. On the other hand, common few-shot models may be weak against noisy negative samples when the number of positive training samples is very small under few-shot conditions. In contrast, our method is designed to deal with noisy negative samples issue, and thus it is more effective. In short, the proposed decoupling classifier is a promising approach to cope with the missing labels issue for FSIS/gFSIS.

Table 2: gFSIS performance for Overall, Base and Novel classes on MS-COCO.

| Shots | Methods | Object Detection | | | | | | Instance Segmentation | | | | | |
|---|---|---|---|---|---|---|---|---|---|---|---|---|---|
| | | Overall | | Base | | Novel | | Overall | | Base | | Novel | |
| | | AP | AP50 | AP | AP50 | AP | AP50 | AP | AP50 | AP | AP50 | AP | AP50 |
| | Base-Only | | | 39.86 | 59.25 | | | | | 32.58 | 55.12 | | |
| 1 | iMTFA [10] | 21.67 | 31.55 | 27.81 | 40.11 | 3.23 | 5.89 | 20.13 | 30.64 | 25.90 | 39.28 | 2.81 | 4.72 |
| | Mask-DeFRCN† [28] | 23.82 | 35.70 | 30.11 | 44.42 | 4.95 | 9.55 | 19.58 | 33.38 | 24.63 | 41.57 | 4.45 | 8.81 |
| | **Ours** | **27.35** | **42.55** | **34.35** | **52.46** | **6.34** | **12.79** | **22.45** | **39.33** | **28.03** | **48.60** | **5.72** | **11.53** |
| 2 | Mask-DeFRCN† [28] | 25.42 | 38.31 | 31.06 | 45.82 | 8.52 | 15.79 | 21.09 | 35.92 | 25.61 | 43.03 | 7.54 | 14.59 |
| | **Ours** | **28.63** | **44.74** | **34.67** | **52.82** | **10.52** | **20.49** | **23.73** | **41.49** | **28.52** | **49.12** | **9.38** | **18.62** |
| 3 | Mask-DeFRCN† [28] | 26.54 | 40.01 | 31.77 | 46.83 | 10.87 | 19.55 | 22.04 | 37.48 | 26.22 | 43.95 | 9.48 | 18.06 |
| | **Ours** | **29.59** | **46.21** | **35.07** | **53.30** | **13.15** | **24.95** | **24.55** | **42.81** | **28.91** | **49.61** | **11.46** | **22.43** |
| 5 | iMTFA [10] | 19.62 | 28.06 | 24.13 | 33.69 | 6.07 | 11.15 | 18.22 | 27.10 | 22.56 | 33.25 | 5.19 | 8.65 |
| | Mask-DeFRCN† [28] | 27.82 | 42.12 | 32.54 | 48.03 | 13.69 | 24.41 | 23.03 | 39.37 | 26.84 | 45.04 | 11.60 | 22.36 |
| | **Ours** | **30.48** | **47.75** | **35.30** | **53.65** | **16.02** | **30.05** | **25.20** | **44.12** | **29.10** | **49.87** | **13.50** | **26.86** |
| 10 | iMTFA [10] | 19.26 | 27.49 | 23.36 | 32.41 | 6.97 | 12.72 | 17.87 | 26.46 | 21.87 | 32.01 | 5.88 | 9.81 |
| | Mask-DeFRCN† [28] | 29.88 | 45.25 | 34.17 | 50.48 | 17.02 | 29.58 | 24.75 | 42.32 | 28.23 | 47.33 | 14.32 | 27.29 |
| | **Ours** | **31.77** | **49.77** | **36.14** | **54.85** | **18.67** | **34.55** | **26.36** | **46.13** | **29.91** | **51.11** | **15.71** | **31.19** |
| 30 | Mask-DeFRCN† [28] | 31.66 | 48.11 | 35.10 | 52.01 | 21.33 | 36.44 | 26.23 | 44.97 | 29.12 | 48.82 | 17.57 | 33.42 |
| | **Ours** | **32.92** | **51.37** | **36.45** | **55.05** | **22.30** | **40.31** | **27.31** | **47.61** | **30.32** | **51.41** | **18.29** | **36.22** |

Table 3: FSOD and gFSOD performance ($AP_{50}$) for Novel classes on PASCAL VOC. The term *w/g* indicates whether we use the gFSOD setting [33]. The superscript ∗ indicates that the results are averaged over 10 times repeated runs on seed0, the same below.

| Methods / Shots | | w/g | Novel Set 1 | | | | | Novel Set 2 | | | | | Novel Set 3 | | | | |
|---|---|---|---|---|---|---|---|---|---|---|---|---|---|---|---|---|---|
| | | | 1 | 2 | 3 | 5 | 10 | 1 | 2 | 3 | 5 | 10 | 1 | 2 | 3 | 5 | 10 |
| FRCN-ft [39] | ICCV 19 | ✗ | 13.8 | 19.6 | 32.8 | 41.5 | 45.6 | 7.9 | 15.3 | 26.2 | 31.6 | 39.1 | 9.8 | 11.3 | 19.1 | 35.0 | 45.1 |
| FSRW [17] | ICCV 19 | ✗ | 14.8 | 15.5 | 26.7 | 33.9 | 47.2 | 15.7 | 15.2 | 22.7 | 30.1 | 40.5 | 21.3 | 25.6 | 28.4 | 42.8 | 45.9 |
| MetaDet [34] | ICCV 19 | ✗ | 18.9 | 20.6 | 30.2 | 36.8 | 49.6 | 21.8 | 23.1 | 27.8 | 31.7 | 43.0 | 20.6 | 23.9 | 29.4 | 43.9 | 44.1 |
| MetaRCNN [39] | ICCV 19 | ✗ | 19.9 | 25.5 | 35.0 | 45.7 | 51.5 | 10.4 | 19.4 | 29.6 | 34.8 | 45.4 | 14.3 | 18.2 | 27.5 | 41.2 | 48.1 |
| TFA [33] | ICML 20 | ✗ | 39.8 | 36.1 | 44.7 | 55.7 | 56.0 | 23.5 | 26.9 | 34.1 | 35.1 | 39.1 | 30.8 | 34.8 | 42.8 | 49.5 | 49.8 |
| MPSR [35] | ECCV 20 | ✗ | 41.7 | - | 51.4 | 55.2 | 61.8 | 24.4 | - | 39.2 | 39.9 | 47.8 | 35.6 | - | 42.3 | 48.0 | 49.7 |
| TIP [19] | CVPR 21 | ✗ | 27.7 | 36.5 | 43.3 | 50.2 | 59.6 | 22.7 | 30.1 | 33.8 | 40.9 | 46.9 | 21.7 | 30.6 | 38.1 | 44.5 | 50.9 |
| DCNet [16] | CVPR 21 | ✗ | 33.9 | 37.4 | 43.7 | 51.1 | 59.6 | 23.2 | 24.8 | 30.6 | 36.7 | 46.6 | 32.3 | 34.9 | 39.7 | 42.6 | 50.7 |
| CME [20] | CVPR 21 | ✗ | 41.5 | 47.5 | 50.4 | 58.2 | 60.9 | 27.2 | 30.2 | 41.4 | 42.5 | 46.8 | 34.3 | 39.6 | 45.1 | 48.3 | 51.5 |
| FSCE [31] | CVPR 21 | ✗ | 44.2 | 43.8 | 51.4 | 61.9 | 63.4 | 27.3 | 29.5 | 43.5 | 44.2 | 50.2 | 37.2 | 41.9 | 47.5 | 54.6 | 58.5 |
| SRR-FSD [43] | CVPR 21 | ✗ | 47.8 | 50.5 | 51.3 | 55.2 | 56.8 | 32.5 | 35.3 | 39.1 | 40.8 | 43.8 | 40.1 | 41.5 | 44.3 | 46.9 | 46.4 |
| FADI [1] | NeurIPS 21 | ✗ | 50.3 | 54.8 | 54.2 | 59.3 | 63.2 | 30.6 | 35.0 | 40.3 | 42.8 | 48.0 | 45.7 | 49.7 | 49.1 | 55.0 | 59.6 |
| FCT [13] | CVPR 22 | ✗ | 38.5 | 49.6 | 53.5 | 59.8 | 64.3 | 25.9 | 34.2 | 40.1 | 44.9 | 47.4 | 34.7 | 43.9 | 49.3 | 53.1 | 56.3 |
| DeFRCN† [28] | ICCV 21 | ✗ | 46.2 | 56.4 | 59.5 | 62.4 | 63.7 | 32.6 | 39.9 | 44.5 | 48.3 | 51.8 | 40.3 | 50.5 | 53.8 | 56.1 | 59.7 |
| **Ours** | | ✗ | **46.2** | **57.4** | **59.9** | **62.9** | **64.5** | **32.6** | **39.9** | 43.4 | 47.9 | 51.3 | 40.3 | 50.5 | 53.8 | **56.9** | **60.7** |
| DeFRCN * [28] | ICCV 21 | ✗ | 53.6 | 57.5 | 61.5 | 64.1 | 60.8 | **30.1** | 38.1 | **47.0** | **53.3** | 47.9 | **48.4** | 50.9 | 52.3 | 54.9 | 57.4 |
| **Ours *** | | ✗ | **56.6** | **59.6** | **62.9** | **65.6** | **62.5** | 29.7 | **38.7** | 46.2 | 48.9 | **48.1** | 47.9 | **51.9** | **53.3** | **56.1** | **59.4** |
| FRCN-ft [39] | ICCV 19 | ✓ | 9.9 | 15.6 | 21.6 | 28.0 | 52.0 | 9.4 | 13.8 | 17.4 | 21.9 | 39.7 | 8.1 | 13.9 | 19.0 | 23.9 | 44.6 |
| FSRW [17] | ICCV 19 | ✓ | 14.2 | 23.6 | 29.8 | 36.5 | 35.6 | 12.3 | 19.6 | 25.1 | 31.4 | 29.8 | 12.5 | 21.3 | 26.8 | 33.8 | 31.0 |
| TFA [33] | ICML 20 | ✓ | 25.3 | 36.4 | 42.1 | 47.9 | 52.8 | 18.3 | 27.5 | 30.9 | 34.1 | 39.5 | 17.9 | 27.2 | 34.3 | 40.8 | 45.6 |
| FSDetView [38] | ECCV 20 | ✓ | 24.2 | 35.3 | 42.2 | 49.1 | 57.4 | 21.6 | 24.6 | 31.9 | 37.0 | 45.7 | 21.2 | 30.0 | 37.2 | 43.8 | 49.6 |
| DeFRCN [28] | ICCV 21 | ✓ | 40.2 | 53.6 | 58.2 | 63.6 | 66.5 | 29.5 | 39.7 | 43.4 | 48.1 | 52.8 | 35.0 | 38.3 | 52.9 | 57.7 | 60.8 |
| **Ours** | | ✓ | **45.8** | **59.1** | **62.1** | **66.8** | **68.0** | **31.8** | **41.7** | **46.6** | **50.3** | **53.7** | **39.6** | **52.1** | **56.3** | **60.3** | **63.3** |

**Few-shot Object Detection on the PASCAL VOC and MS-COCO.** *Our method significantly outperforms the state-of-the-art few-shot object detection methods by a large margin both on the PASCAL VOC and MS-COCO datasets under gFSOD setting again. For the FSOD setting, our method is also better than the state-of-the-art under most cases.* The results on the PASCAL VOC and MS-COCO are reported Tables 3, 5 and 4, respectively. Some interesting observations are summarized as follows: **1)** Our method significantly and consistently exceeds the current state-of-the-art DeFRCN under the gFSOD setting both on the PASCAL VOC and MS-COCO, which is similar to that of the gF-SIS; **2)** Our method is better than the strong DeFRCN in all shots on the MS-COCO and in most cases on the PASCAL VOC under the FSOD setting (averaging the results on 10 seeds). It is worth noting that our method's performance on the MS-COCO is close (maybe slightly worse, about 0.5%) to that of DeFRCN if we compare the results based on 10 times repeated runs on the seed0. We recheck the missing rate of the seed0, and find

Table 5: FSOD performance (AP) for Novel classes on MS-COCO. The superscripts x indicate that the results are reported in DeFRCN [28].

| Methods / Shots | | 1 | 2 | 3 | 5 | 10 | 30 |
|---|---|---|---|---|---|---|---|
| FRCN-ft [39] | ICCV 19 | 1.0 | 1.8 | 2.8 | 4.0 | 6.5 | 11.1 |
| FSRW [17] | ICCV 19 | - | - | - | - | 5.6 | 9.1 |
| MetaDet [34] | ICCV 19 | - | - | - | - | 7.1 | 11.3 |
| MetaRCNN [39] | ICCV 19 | - | - | - | - | 8.7 | 12.4 |
| TFA [33] | ICML 20 | 4.4 | 5.4 | 6.0 | 7.7 | 10.0 | 13.7 |
| MPSR [35] | ECCV 20 | 5.1 | 6.7 | 7.4 | 8.7 | 9.8 | 14.1 |
| FSDetView [38] | ICCV 20 | 4.5 | 6.6 | 7.2 | 10.7 | 12.5 | 14.7 |
| TIP [19] | CVPR 21 | - | - | - | - | 16.3 | 18.3 |
| DCNet [16] | CVPR 21 | - | - | - | - | 12.8 | 18.6 |
| CME [20] | CVPR 21 | - | - | - | - | 15.1 | 16.9 |
| FSCE [31] | CVPR 21 | - | - | - | - | 11.1 | 15.3 |
| SRR-FSD [43] | CVPR 21 | - | - | - | - | 11.3 | 14.7 |
| FADI [1] | NeurIPS 21 | 5.7 | 7.0 | 8.6 | 10.1 | 12.2 | 16.1 |
| FCT [13] | CVPR 22 | 5.1 | 7.2 | 9.8 | 12.0 | 15.3 | 20.2 |
| DeFRCN† [28] | ICCV 21 | 7.7 | 11.4 | 13.3 | 15.5 | 18.5 | 22.5 |
| **Ours** | | **8.1** | **12.1** | **14.4** | **16.6** | **19.5** | **22.7** |
| DeFRCN * [28] | ICCV 21 | 9.3 | 12.9 | 14.8 | 16.1 | 18.5 | 22.6 |
| **Ours *** | | **10.0** | **13.6** | 14.7 | 15.7 | 18.0 | 22.2 |

setting (averaging the results on 10 seeds). It is worth noting that our method's performance on the MS-COCO is close (maybe slightly worse, about 0.5%) to that of DeFRCN if we compare the results based on 10 times repeated runs on the seed0. We recheck the missing rate of the seed0, and find

Table 4: gFSOD performance (AP) for **O**verall, **B**ase and **N**ovel classes on MS-COCO.

| Method / Shots | 1 | | | 2 | | | 3 | | | 5 | | | 10 | | | 30 | | |
|---|---|---|---|---|---|---|---|---|---|---|---|---|---|---|---|---|---|---|
| | O | B | N | O | B | N | O | B | N | O | B | N | O | B | N | O | B | N |
| FRCN-ft [39] | 16.2 | 21.0 | 1.7 | 15.8 | 20.0 | 3.1 | 15.0 | 18.8 | 3.7 | 14.4 | 17.6 | 4.6 | 13.4 | 16.1 | 5.5 | 13.5 | 15.6 | 7.4 |
| TFA [33] | 24.4 | 31.9 | 1.9 | 24.9 | 31.9 | 3.9 | 25.3 | 32.0 | 5.1 | 25.9 | 41.2 | 7.0 | 26.6 | 32.4 | 9.1 | 28.7 | 34.2 | 12.1 |
| FSDetView [38] | | | 3.2 | | | 4.9 | | | 6.7 | | | 8.1 | | | 10.7 | | | 15.9 |
| DeFRCN [28] | 24.4 | 30.4 | 4.8 | 25.7 | 31.4 | 8.5 | 26.6 | 32.1 | 10.7 | 27.8 | 32.6 | 13.6 | 29.7 | 34.0 | 16.8 | 31.4 | 34.8 | 21.2 |
| **Ours** | **27.4** | **34.4** | **6.2** | **28.6** | **34.7** | **10.4** | **29.4** | **34.9** | **12.9** | **30.2** | **35.0** | **15.7** | **31.4** | **35.7** | **18.3** | **32.3** | **35.8** | **21.9** |

Table 6: The effects of DC and PCB for gFSIS performance on MS-COCO. GFLOPs are averaged over all 5000 MS-COCO validation images.

| Shots | M-Rate | DC | PCB | Complexity | | Detection | | | | Segmentation | | | |
|---|---|---|---|---|---|---|---|---|---|---|---|---|---|
| | | | | #Params. | GFLOPs | Base | | Novel | | Base | | Novel | |
| | | | | | | AP | AP50 | AP | AP50 | AP | AP50 | AP | AP50 |
| 1 | 83.3% | ✗ | ✗ | 54.9M | 334.54 | 30.09 | 44.45 | 3.89 | 7.43 | 24.62 | 41.58 | 3.52 | 6.88 |
| | | ✓ | ✗ | 54.9M | 334.54 | 34.35 | 52.46 | 5.04 | 10.03 | 28.03 | 48.60 | 4.59 | 9.12 |
| | | ✗ | ✓ | 99.4M | 377.88 | 30.11 | 44.42 | 4.95 | 9.55 | 24.63 | 41.57 | 4.45 | 8.81 |
| | | ✓ | ✓ | 99.4M | 377.88 | **34.35** | **52.46** | **6.34** | **12.79** | **28.03** | **48.60** | **5.72** | **11.53** |
| 5 | 80.3% | ✗ | ✗ | 54.9M | 334.54 | 32.54 | 48.03 | 11.94 | 21.16 | 26.84 | 45.04 | 10.10 | 19.37 |
| | | ✓ | ✗ | 54.9M | 334.54 | 35.30 | 53.65 | 14.01 | 26.17 | 29.10 | 49.87 | 11.80 | 23.38 |
| | | ✗ | ✓ | 99.4M | 377.88 | 32.54 | 48.03 | 13.69 | 24.41 | 26.84 | 45.04 | 11.60 | 22.36 |
| | | ✓ | ✓ | 99.4M | 377.88 | **35.30** | **53.65** | **16.02** | **30.05** | **29.10** | **49.87** | **13.50** | **26.86** |
| 10 | 76.7% | ✗ | ✗ | 54.9M | 334.54 | 34.05 | 50.21 | 14.96 | 25.70 | 28.12 | 47.10 | 12.60 | 23.81 |
| | | ✓ | ✗ | 54.9M | 334.54 | 36.13 | 54.81 | 16.66 | 30.79 | 29.90 | 51.07 | 13.98 | 27.72 |
| | | ✗ | ✓ | 99.4M | 377.88 | 34.17 | 50.48 | 17.02 | 29.58 | 28.23 | 47.33 | 14.32 | 27.29 |
| | | ✓ | ✓ | 99.4M | 377.88 | **36.14** | **54.85** | **18.67** | **34.55** | **29.91** | **51.11** | **15.71** | **31.19** |

that the corresponding missing rate is significantly lowered (even zero) than that of the other 9 seeds. This also further indicates that our method is robust when the missing rate is small even zero.

## 4.3 Ablation Study and Analysis

We conduct the ablation study to analyze the component of our method. Models in this section are based on the gFSIS setting (1-, 5-, 10-shot) using MS-COCO. Note that the DeFRCN uses a Prototypical Calibration Block (PCB) to refine the classification score which is effective for improving the FSOD performance, but this brings additional computation cost. Therefore, we consider these two factors including decoupling classifier (DC) head and PCB in the following analysis.

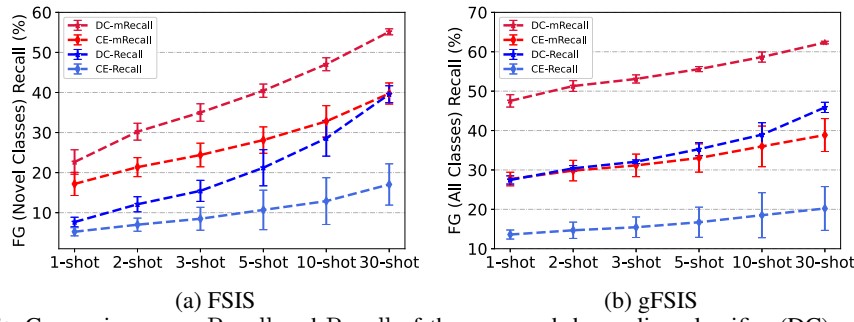

(a) FSIS  (b) gFSIS

Figure 3: Comparison on mRecall and Recall of the proposed decoupling classifier (DC) and standard classification head (CE) under FSIS and gFSIS settings. The mean and standard deviation results are computed on all 10 seeds for each shot. Best viewed in color and zoom in.

**Effectiveness.** We only simply replace the standard classifier with the proposed decoupling classifier in DeFRCN, which results in significant improvements, especially for the higher missing rate (e.g., low shot on MS-COCO). What's more, our decoupling classifier is effective not only on novel classes but also on base classes, while the PCB seems only effective on novel classes. In addition, our decoupling classifier without the PCB significantly outperforms the counterpart with the PCB on base classes and is also comparable on novel classes.

**Efficiency.** Firstly, our decoupling classifier does not introduce any additional parameters or computation cost. Secondly, our method obtains better detection and segmentation performance when using the same complexity as Mask-DeFRCN whether the PCB is used or not. Last but not least, we only need almost half of the parameters and fewer GFLOPs when removing the PCB block, and still achieve significant improvements on base classes and comparable performance on novel classes compared to Mask-DeFRCN using the PCB.

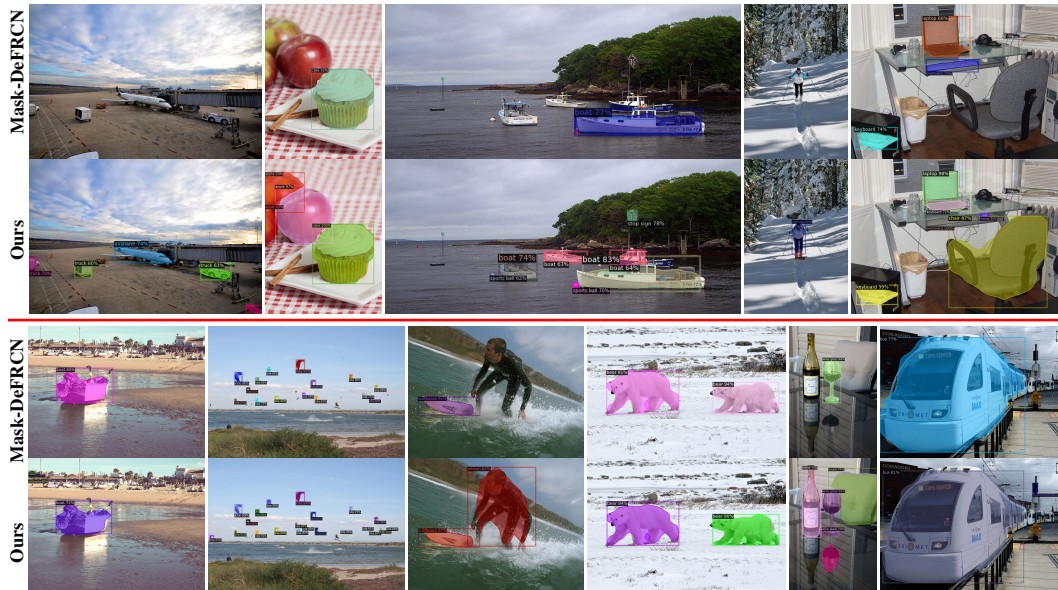

Figure 4: Visualization results of our method and the strong baseline (Mask-DeFRCN) on MS-COCO validation images. Best viewed in color and zoom in.

**Why DC works?** We have given some analysis from the perspective of gradient optimization in Sec. 3. Here, we try to discuss from the generalization ability of decoupling classifier and compare with the baseline. We want to explore whether the decoupling classifier mitigates the bias classification. To this end, we employ $\mathrm{Recall}$ metric to evaluate the classification head for all ground-truth foreground objects. Note that the classification head outputs a multi-class probability distribution $\hat{\boldsymbol{p}} \in R^{C+1}$. The predicted class is determined by $\mathrm{argmax}_i \hat{p}_i$. We define that an object is recalled if its prediction is not background, i.e., any foreground category. Considering that the number of each foreground category varies considerably, we also compare $\mathrm{mRecall}$ (mean $\mathrm{Recalls}$ of all classes). The comparison results are shown in Fig. 3. We can see that the $\mathrm{mRecall}$ and $\mathrm{Recall}$ of the decoupling classifier significantly outperforms the standard one on each shot both FSIS and gFSIS. This indicates that our decoupling classifier is helpful to mitigate the bias classification thus boosting the performance of FSIS and gFSIS .

**Qualitative Evaluation** In Fig. 4, we visualize the results of our method and the strong baseline (Mask-DeFRCN) on MS-COCO validation images with 10-shot setting for gFSIS task. In the top rows, we show success cases with our method but partly failures with the baseline. These failures are mainly caused by the missing detection because the baseline method may tend to incorrectly recognize positive objects as background (i.e., bias classification). In addition, our method may also produce some failure predictions as shown in the bottom row from left to right, including the missing detection of small or occlusion objects, coarse boundary segmentation, and the misclassification of similar appearance objects.

## 5   Conclusion

In this paper, we firstly find that the missing label widely exists in few-shot scenario. Furthermore, we analyze that the missing label issue may result in biased classification and thus limit the generalization ability on novel classes. Therefore, we propose a simple but effective method that decouples the standard classifier into two parallel heads to independently process positive and negative examples. Comprehensive experiments on the few-shot object detection and instance segmentation benchmark datasets show that our approach can effectively and efficiently boost FSOD/gFSOD and FSIS/gFSIS performance without any additional parameters and computation cost. We hope this study attract more interest in designing a simple method for FSOD or FSIS in the future. A limitation of our method is that it may not be suitable when the missing label rate is small. However, our method is still comparable to its counterpart even if the missing label rate is zero, which indicates its robustness.

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
