| MTFA [10] | *CVPR 21* | | 2.47 | 4.85 | - | - | - | - | 6.61 | 12.32 | 8.52 | 15.53 | - | - |
| iMTFA [10] | *CVPR 21* | | 3.28 | 6.01 | - | - | - | - | 6.22 | 11.28 | 7.14 | 12.91 | - | - |
| Mask-DeFRCN$^\dagger$ [28] | *ICCV 21* | | 7.54 | 14.46 | 11.01 | 20.20 | 13.07 | 23.28 | 15.39 | 27.29 | 18.72 | 32.80 | 22.63 | 38.95 |
| **Ours** | | | **8.09** | **15.85** | **11.90** | **22.39** | **14.04** | **25.74** | **16.39** | **29.96** | **19.33** | **34.78** | **22.73** | **40.24** |
| Meta R-CNN [39] | *ICCV 19* | Seg | - | - | - | - | - | - | 2.8 | 6.9 | 4.4 | 10.6 | - | - |
| MTFA [10] | *CVPR 21* | | 2.66 | 4.56 | - | - | - | - | 6.62 | 11.58 | 8.39 | 14.64 | - | - |
| iMTFA [10] | *CVPR 21* | | 2.83 | 4.75 | - | - | - | - | 5.24 | 8.73 | 5.94 | 9.96 | - | - |

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

## Appendix: Supplementary Material

In this supplementary material, we first give the training details about the base pre-training and novel fine-tuning of our method in Sec. A. Then, we provide the PyTorch-like style codes for our decoupling classifier in Sec. B. Next, we provide complete results including average and standard deviation of multiple runs on PASCAL VOC and MS-COCO for FSOD/FSIS and gFSOD/gFSIS in Sec. C. Furthermore, more visualized results of our method and the strong baseline (Mask-DeFRCN) on MS-COCO validation images are showed in Sec. D. Finally, we include the missing rates on few-shot PASCAL VOC and MS-COCO in Sec. E.

## A   Training Details

Following the two-stage training procedure of TFA [33] and DeFRCN [28], we first pre-train model with abundant labeled images for base classes and then fine-tune the model with few-shot labeled images for novel classes or base-novel classes. For the first stage training, we employ the standard classifier (i.e., cross entropy loss) because all base class objects are completely labeled. In the second stage, we only simply replace the standard classifier with the proposed decoupling classifier for mitigating the bias classification under few-shot setting. For a fair comparison, we use the same hyper-parameters in the DeFRCN [28], such as batch size, learning rate, and training iterations.

## B   The Core Code for Decoupling Classifier

---
**Algorithm 1** PyTorch-like Style Code for Decoupling Classifier.

---

```python
def dc_loss(x, y, m):
    """
    Compute loss for the decoupling classifier.
    Return scalar Tensor for single image.

    Args:
        x: predicted class scores in [-inf, +inf], x's size: N x (1+C), where N is the
            number of region proposals of one image.
        y: ground-truth classification labels in [0, C-1], y's size: N x 1, where [0,C-1]
            represent foreground classes and C-1 represents the background class.
        m: image-level label vector and its element is 0 or 1, m's size: 1 x (1+C)

    Returns:
        loss
    """

    # background class index
    N = x.shape[0]
    bg_label = x.shape[1]-1

    # positive head
    pos_ind = y!=bg_label
    pos_logit = x[pos_ind,:]
    pos_score = F.softmax(pos_logit, dim=1) # Eq. 4
    pos_loss = F.nll_loss(pos_score.log(), y[pos_ind], reduction="sum") #Eq. 5

    # negative head
    neg_ind = y==bg_label
    neg_logit = x[neg_ind,:]
    neg_score = F.softmax(m.expand_as(neg_logit)*neg_logit, dim=1) #Eq. 8
    neg_loss = F.nll_loss(neg_score.log(), y[neg_ind], reduction="sum")  #Eq. 9

    # total loss
    loss = (pos_loss + neg_loss)/N #Eq. 6

    return loss
```

---

Algorithm 1 provides the PyTorch-like style code for our decoupling classifier. It can be seen that it is very simple (core implementation only uses one line of code, the main change is to only introduce an image-level label vector, $m$ in Eq. 8, into the standard softmax function for the negative head and keep others unchanged like the positive head) but really effective (e.g., 5.6 AP50 improvements for detection and 4.5 AP50 improvements for segmentation on challenging MS-COCO with 5-shot setting in Table 8).

## C  Complete Results of FSOD/FSIS and gFSOD/gFSIS

In our main paper, we only report the average AP/AP50 metric for FSOD/FSIS and gFSOD/gFSIS on MS-COCO and PASCAL VOC datasets. In this supplementary material, we report the average AP/AP50 metric with 95% confidence interval over 10 seeds for FSOD/FSIS and gFSOD/gFSIS in Tables 7, 8, 9, 10 and 11, respectively. $K=\{1, 2, 3, 5, 10, 30\}$ is the number of labeled instances of each class used in the fine-tuning stage.

Table 7: FSIS performance (AP and AP50) for Novel classes on MS-COCO. Note that the superscript † indicates that the results are our re-implementation, the red numers indicate the performance improvements of our method compared to the baseline, and the best results are in bold, the same below.

| Methods | Tasks | 1 AP | 1 AP50 | 2 AP | 2 AP50 | 3 AP | 3 AP50 | 5 AP | 5 AP50 | 10 AP | 10 AP50 | 30 AP | 30 AP50 |
|---|---|---|---|---|---|---|---|---|---|---|---|---|---|
| Mask-DeFRCN† [28] | Det | 7.54±0.5 | 14.46±0.9 | 11.01±0.5 | 20.20±0.7 | 13.07±0.6 | 23.28±1.0 | 15.39±0.7 | 27.29±1.0 | 18.72±0.3 | 32.80±0.6 | 22.63±0.3 | 38.95±0.5 |
| Ours | | **8.09**±0.4 | **15.85**±0.8 | **11.90**±0.4 | **22.39**±0.7 | **14.04**±0.6 | **25.74**±0.9 | **16.39**±0.6 | **29.96**±0.9 | **19.33**±0.4 | **34.78**±0.8 | **22.73**±0.4 | **40.24**±0.6 |
| | | +0.55 | +1.39 | +0.89 | +2.19 | +0.97 | +2.46 | +1.00 | +2.67 | +0.61 | +1.98 | +0.10 | +1.29 |
| Mask-DeFRCN† [28] | Seg | 6.69±0.5 | 13.24±0.8 | 9.51±0.5 | 18.58±0.7 | 11.01±0.4 | 21.27±0.9 | 12.66±0.6 | 24.58±1.0 | 15.39±0.3 | 29.71±0.6 | 18.28±0.3 | 35.20±0.5 |
| Ours | | **7.18**±0.5 | **14.33**±0.8 | **10.31**±0.4 | **20.43**±0.7 | **11.85**±0.4 | **23.24**±0.8 | **13.48**±0.5 | **26.67**±0.9 | **15.85**±0.4 | **31.33**±0.7 | **18.34**±0.3 | **35.99**±0.6 |
| | | +0.49 | +1.09 | +0.80 | +1.85 | +0.84 | +1.97 | +0.82 | +2.09 | +0.46 | +1.62 | +0.06 | +0.79 |

Table 8: gFSIS performance (AP and AP50) for Overall, Base and Novel classes on MS-COCO.

| Shots | Methods | Object Detection Overall #80 AP | AP50 | Base #60 AP | AP50 | Novel #20 AP | AP50 | Instance Segmentation Overall #80 AP | AP50 | Base #60 AP | AP50 | Novel #20 AP | AP50 |
|---|---|---|---|---|---|---|---|---|---|---|---|---|---|
| | Base-Only | 39.86 | 59.25 | | | | | 32.58 | 55.12 | | | | |
| 1 | Mask-DeFRCN† [28] | 23.82±0.5 | 35.70±0.7 | 30.11±0.6 | 44.42±0.9 | 4.95±0.4 | 9.55±0.7 | 19.58±0.4 | 33.38±0.7 | 24.63±0.5 | 41.57±0.9 | 4.45±0.5 | 8.81±0.7 |
| | Ours | **27.35**±0.3 | **42.55**±0.3 | **34.35**±0.3 | **52.46**±0.3 | **6.34**±0.4 | **12.79**±0.9 | **22.45**±0.2 | **39.33**±0.3 | **28.03**±0.2 | **48.60**±0.3 | **5.72**±0.5 | **11.53**±0.9 |
| | | +3.53 | +6.85 | +4.24 | +8.04 | +1.39 | +3.24 | +2.87 | +6.45 | +3.40 | +7.03 | +1.27 | +2.72 |
| 2 | Mask-DeFRCN† [28] | 25.42±0.5 | 38.31±0.8 | 31.06±0.5 | 45.82±0.7 | 8.52±0.8 | 15.79±1.1 | 21.09±0.4 | 35.92±0.8 | 25.61±0.3 | 43.03±0.7 | 7.54±0.8 | 14.59±1.1 |
| | Ours | **28.63**±0.3 | **44.74**±0.5 | **34.67**±0.3 | **52.82**±0.4 | **10.52**±0.7 | **20.49**±1.1 | **23.73**±0.3 | **41.49**±0.4 | **28.52**±0.2 | **49.12**±0.3 | **9.38**±0.7 | **18.62**±1.1 |
| | | +3.21 | +6.43 | +3.61 | +7.00 | +2.00 | +4.70 | +2.64 | +5.57 | +2.91 | +6.09 | +1.84 | +4.03 |
| 3 | Mask-DeFRCN† [28] | 26.54±0.5 | 40.01±0.7 | 31.77±0.4 | 46.83±0.6 | 10.87±0.8 | 19.55±1.2 | 22.04±0.4 | 37.48±0.7 | 26.22±0.3 | 43.95±0.5 | 9.48±0.7 | 18.06±1.1 |
| | Ours | **29.59**±0.2 | **46.21**±0.4 | **35.07**±0.2 | **53.30**±0.4 | **13.15**±0.5 | **24.95**±0.8 | **24.55**±0.2 | **42.81**±0.3 | **28.91**±0.2 | **49.61**±0.4 | **11.46**±0.4 | **22.43**±0.8 |
| | | +3.05 | +6.20 | +3.30 | +6.47 | +2.28 | +5.40 | +2.51 | +5.33 | +2.69 | +5.66 | +1.98 | +4.37 |
| 5 | Mask-DeFRCN† [28] | 27.82±0.4 | 42.12±0.6 | 32.54±0.4 | 48.03±0.5 | 13.69±0.7 | 24.41±1.3 | 23.03±0.3 | 39.37±0.6 | 26.84±0.3 | 45.04±0.5 | 11.60±0.7 | 22.36±1.2 |
| | Ours | **30.48**±0.2 | **47.75**±0.3 | **35.30**±0.2 | **53.65**±0.3 | **16.02**±0.5 | **30.05**±0.8 | **25.20**±0.2 | **44.12**±0.3 | **29.10**±0.2 | **49.87**±0.3 | **13.50**±0.5 | **26.86**±0.9 |
| | | +2.66 | +5.63 | +2.76 | +5.62 | +2.33 | +5.64 | +2.17 | +4.75 | +2.26 | +4.83 | +1.90 | +4.50 |
| 10 | Mask-DeFRCN† [28] | 29.88±0.3 | 45.25±0.7 | 34.17±0.5 | 50.48±0.5 | 17.02±0.6 | 29.58±1.2 | 24.75±0.3 | 42.32±0.6 | 28.23±0.2 | 47.33±0.5 | 14.32±0.6 | 27.29±1.1 |
| | Ours | **31.77**±0.2 | **49.77**±0.3 | **36.14**±0.2 | **54.85**±0.2 | **18.67**±0.4 | **34.55**±0.7 | **26.36**±0.2 | **46.13**±0.3 | **29.91**±0.2 | **51.11**±0.2 | **15.71**±0.4 | **31.19**±0.7 |
| | | +1.89 | +4.52 | +1.97 | +4.37 | +1.65 | +4.97 | +1.61 | +3.81 | +1.68 | +3.78 | +1.39 | +3.90 |
| 30 | Mask-DeFRCN† [28] | 31.66±0.1 | 48.11±0.2 | 35.10±0.1 | 52.01±0.2 | 21.33±0.4 | 36.44±0.7 | 26.23±0.1 | 44.97±0.2 | 29.12±0.1 | 48.82±0.1 | 17.57±0.4 | 33.42±0.7 |
| | Ours | **32.92**±0.2 | **51.37**±0.4 | **36.45**±0.3 | **55.05**±0.4 | **22.30**±0.4 | **40.31**±0.6 | **27.31**±0.2 | **47.61**±0.4 | **30.32**±0.2 | **51.41**±0.4 | **18.29**±0.3 | **36.22**±0.6 |
| | | +1.26 | +3.26 | +1.35 | +3.04 | +0.97 | +3.87 | +1.08 | +2.64 | +1.20 | +2.59 | +0.72 | +2.80 |

Table 9: FSOD performance (AP and AP50) for Novel classes on MS-COCO.

| Methods / Shots | | 1 AP | 1 AP50 | 2 AP | 2 AP50 | 3 AP | 3 AP50 | 5 AP | 5 AP50 | 10 AP | 10 AP50 | 30 AP | 30 AP50 |
|---|---|---|---|---|---|---|---|---|---|---|---|---|---|
| DeFRCN† [28] | ICCV 21 | 7.7±0.6 | 15.1±0.9 | 11.4±0.5 | 21.4±0.8 | 13.3±0.4 | 24.5±0.9 | 15.5±0.5 | 28.3±0.9 | 18.5±0.4 | 33.4±0.6 | 22.5±0.3 | 39.5±0.4 |
| Ours | | **8.1**±0.6 | **16.3**±0.9 | **12.1**±0.5 | **23.4**±0.7 | **14.4**±0.4 | **27.1**±1.0 | **16.6**±0.5 | **31.1**±0.9 | **19.5**±0.5 | **35.8**±0.8 | **22.7**±0.4 | **41.0**±0.6 |
| | | +0.4 | +1.2 | +0.7 | +2.0 | +1.1 | +2.6 | +1.1 | +2.8 | +1.0 | +2.4 | +0.2 | +0.5 |

## D  Qualitative Evaluation

In Fig. 6, we visualize the results of our method and the strong baseline (Mask-DeFRCN) on MS-COCO validation images using the gFSIS setting with $K=10$. The

Table 10: gFSOD performance (AP and AP50) for Overall, Base and Novel classes on MS-COCO.

| # shots | Methods | | Overall #80 | | | Base #60 | Novel #20 | |
|---|---|---|---|---|---|---|---|---|
| | | | AP | AP50 | AP75 | AP | AP | AP50 |
| 1 | FRCN+ft [39] | ICCV 19 | 16.2±0.9 | 25.8±1.2 | 17.6±1.0 | 21.0±1.2 | 1.7±0.2 | 3.3 |
| | TFA [33] | ICML 20 | 24.4±0.6 | 39.8±0.8 | 26.1±0.8 | 31.9±0.7 | 1.9±0.4 | 3.8 |
| | DeFRCN [28] | ICCV 21 | 24.0±0.4 | 36.9±0.6 | 26.2±0.4 | 30.4±0.4 | 4.8±0.6 | 9.5±0.9 |
| | **Ours** | | **27.4±0.2** +3.4 | **43.4±0.4** +6.5 | **29.4±0.3** +3.2 | **34.4±0.3** +4.0 | **6.2±0.6** +1.4 | **12.7±0.9** +3.2 |
| 2 | FRCN+ft [39] | ICCV 19 | 15.8±0.7 | 25.0±1.1 | 17.3±0.7 | 20.0±0.9 | 3.1±0.3 | 6.1 |
| | TFA [33] | ICML 20 | 24.9±0.6 | 40.1±0.9 | 27.0±0.7 | 31.9±0.7 | 3.9±0.4 | 7.8 |
| | DeFRCN [28] | ICCV 21 | 25.7±0.5 | 39.6±0.8 | 28.0±0.5 | 31.4±0.4 | 8.5±0.8 | 16.3±1.4 |
| | **Ours** | | **28.6±0.3** +2.9 | **45.6±0.5** +6.0 | **30.7±0.4** +2.7 | **34.7±0.3** +3.3 | **10.4±0.8** +1.9 | **20.9±1.3** +4.6 |
| 3 | FRCN+ft [39] | ICCV 19 | 15.0±0.7 | 23.9±1.2 | 16.4±0.7 | 18.8±0.9 | 3.7±0.4 | 7.1 |
| | TFA [33] | ICML 20 | 25.3±0.6 | 40.4±1.0 | 27.6±0.7 | 32.0±0.7 | 5.1±0.6 | 9.9 |
| | DeFRCN [28] | ICCV 21 | 26.6±0.4 | 41.1±0.7 | 28.9±0.4 | 32.1±0.3 | 10.7±0.8 | 20.0±1.2 |
| | **Ours** | | **29.4±0.2** +2.8 | **46.8±0.3** +5.7 | **31.4±0.3** +2.5 | **34.9±0.2** +2.8 | **12.9±0.6** +2.2 | **25.1±1.0** +5.1 |
| 5 | FRCN+ft [39] | ICCV 19 | 14.4±0.8 | 23.0±1.3 | 15.6±0.8 | 17.6±0.9 | 4.6±0.5 | 8.7 |
| | TFA [33] | ICML 20 | 25.9±0.6 | 41.2±0.9 | 28.4±0.6 | 32.3±0.6 | 7.0±0.7 | 13.3 |
| | DeFRCN [28] | ICCV 21 | 27.8±0.3 | 43.0±0.6 | 30.2±0.3 | 32.6±0.3 | 13.6±0.7 | 24.7±1.1 |
| | **Ours** | | **30.2±0.2** +2.4 | **48.2±0.3** +5.2 | **32.2±0.2** +2.0 | **35.0±0.2** +3.6 | **15.7±0.5** +2.1 | **30.3±0.9** +5.6 |
| 10 | FRCN+ft [39] | ICCV 19 | 13.4±1.0 | 21.8±1.7 | 14.5±0.9 | 16.1±1.0 | 5.5±0.9 | 10.0 |
| | TFA [33] | ICML 20 | 26.6±0.5 | 42.2±0.8 | 29.0±0.6 | 32.4±0.6 | 9.1±0.5 | 17.1 |
| | DeFRCN [28] | ICCV 21 | 29.7±0.2 | 46.0±0.5 | 32.1±0.2 | 34.0±0.2 | 16.8±0.6 | 29.6±1.3 |
| | **Ours** | | **31.4±0.2** +1.7 | **49.9±0.3** +3.9 | **33.4±0.2** +1.3 | **35.7±0.2** +1.7 | **18.3±0.4** +1.5 | **34.5±0.6** +4.9 |
| 30 | FRCN+ft [39] | ICCV 19 | 13.5±1.0 | 21.8±1.9 | 14.5±1.0 | 15.6±1.0 | 7.4±1.1 | 13.1 |
| | TFA [33] | ICML 20 | 28.7±0.4 | 44.7±0.7 | 31.5±0.4 | 34.2±0.4 | 12.1±0.4 | 22.0 |
| | DeFRCN [28] | ICCV 21 | 31.4±0.1 | 48.8±0.2 | 33.9±0.1 | 34.8±0.1 | 21.2±0.4 | 36.7±0.8 |
| | **Ours** | | **32.3±0.2** +0.9 | **51.3±0.3** +2.5 | **34.5±0.2** +0.6 | **35.8±0.2** +1.0 | **21.9±0.3** +0.7 | **40.2±0.5** +3.5 |

top rows show success cases while the bottom row shows failure cases. In the middle rows, we show success cases with our method but partly failures with the baseline. These failures are mainly caused by the missing detection because the baseline method may tend to incorrectly recognize positive objects as background (i.e., bias classification). In addition, our method may also produce failure predictions as shown in the bottom row from left to right, including the missing detection of small or occultation objects, coarse boundary segmentation, and the misclassification of similar appearance objects.

# E   The Proportion of Missing Labeled Instances

Here, we provide the detailed missing rates on each seed for MS-COCO in Fig. 5 and PASCAL VOC in Fig. 7.

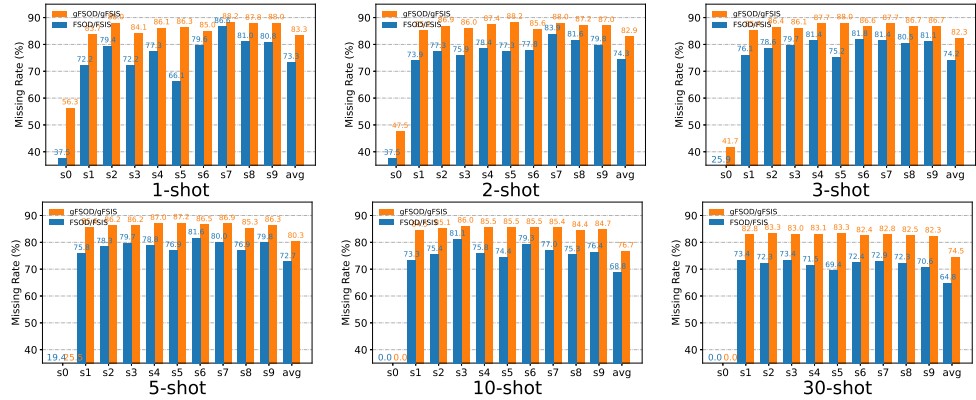

Figure 5: Comparisons of the proportion of missing labeled instances of FSOD/FSIS and gFSOD/gFSIS on the MS-COCO dataset. We can see that there are high proportions almost on all shots using different seeds except the seed0; and the gFSOD/gFSIS setting generally has higher missing rates than that of FSOD/FSIS.

Table 11: gFSOD performance (AP and AP50) on PASCAL VOC dataset.

| Set | # shots | Method | Overall #20 | | | Base #15 | Novel #5 | |
|---|---|---|---|---|---|---|---|---|
| | | | AP | AP50 | AP75 | AP | AP | AP50 |
| Set 1 | 1 | FSRW [17] | 27.6±0.5 | 50.8±0.9 | 26.5±0.6 | 34.1±0.5 | 8.0±1.0 | 14.2 |
| | | FRCN+ft [39] | 30.2±0.6 | 49.4±0.7 | 32.2±0.9 | 38.2±0.8 | 6.0±0.7 | 9.9 |
| | | TFA [33] | 40.6±0.5 | 64.5±0.6 | 44.7±0.6 | 49.4±0.4 | 14.2±1.4 | 25.3 |
| | | DeFRCN [28] | 42.0±0.6 | 66.7±0.8 | 45.5±0.7 | 48.4±0.4 | 22.5±1.7 | 40.2 |
| | | **Ours** | **43.5**±0.9 **+1.5** | **69.7**±1.4 **+3.0** | **47.2**±1.1 **+1.7** | **49.5**±0.7 **+1.1** | **25.6**±2.6 **+3.1** | **45.8**±4.5 **+5.6** |
| | 2 | FSRW [17] | 28.7±0.4 | 52.2±0.6 | 27.7±0.5 | 33.9±0.4 | 13.2±1.0 | 23.6 |
| | | FRCN+ft [39] | 30.5±0.6 | 49.4±0.8 | 32.6±0.7 | 37.3±0.7 | 9.9±0.9 | 15.6 |
| | | TFA [33] | 42.6±0.3 | 67.1±0.4 | 47.0±0.4 | 49.6±0.3 | 21.7±1.0 | 36.4 |
| | | DeFRCN [28] | 44.3±0.4 | 70.2±0.5 | 48.0±0.6 | 49.1±0.3 | 30.6±1.2 | 53.6 |
| | | **Ours** | **45.6**±0.5 **+1.3** | **73.2**±0.8 **+3.0** | **49.2**±0.8 **+1.2** | **49.7**±0.4 **+0.6** | **33.4**±1.6 **+2.8** | **59.1**±2.7 **+5.5** |
| | 3 | FSRW [17] | 29.5±0.3 | 53.3±0.6 | 28.6±0.4 | 33.8±0.3 | 16.8±0.9 | 29.8 |
| | | FRCN+ft [39] | 31.8±0.5 | 51.4±0.8 | 34.2±0.6 | 37.9±0.5 | 13.7±1.0 | 21.6 |
| | | TFA [33] | 43.7±0.3 | 68.5±0.4 | 48.3±0.4 | 49.8±0.3 | 25.4±0.9 | 42.1 |
| | | DeFRCN [28] | 45.3±0.3 | 71.5±0.4 | 49.0±0.5 | 49.3±0.3 | 33.7±0.8 | 58.2 |
| | | **Ours** | **46.4**±0.6 **+1.1** | **74.1**±0.6 **+2.6** | **50.1**±0.8 **+1.1** | **50.0**±0.5 **+0.7** | **35.5**±1.6 **+1.8** | **62.1**±2.1 **+3.9** |
| | 5 | FSRW [17] | 30.4±0.3 | 54.6±0.5 | 29.6±0.4 | 33.7±0.3 | 20.6±0.8 | 36.5 |
| | | FRCN+ft [39] | 32.7±0.5 | 52.5±0.8 | 35.0±0.6 | 37.6±0.4 | 17.9±1.1 | 28.0 |
| | | TFA [33] | 44.8±0.3 | 70.1±0.4 | 49.4±0.4 | 50.1±0.2 | 28.9±0.8 | 47.9 |
| | | DeFRCN [28] | 46.4±0.3 | 73.1±0.3 | 50.4±0.4 | 49.6±0.3 | 37.3±0.8 | 63.6 |
| | | **Ours** | **47.5**±0.5 **+1.1** | **75.3**±0.4 **+2.2** | **51.4**±0.6 **+1.0** | **50.4**±0.4 **+0.8** | **38.6**±0.8 **+1.3** | **66.8**±0.8 **+3.2** |
| | 10 | FRCN+ft [33] | 33.3±0.4 | 53.8±0.6 | 35.5±0.4 | 36.8±0.4 | 22.7±0.9 | 52.0 |
| | | TFA [33] | 45.8±0.2 | 71.3±0.3 | 50.4±0.3 | 50.4±0.2 | 32.0±0.6 | 52.8 |
| | | DeFRCN [28] | 47.2±0.2 | 74.0±0.3 | 51.3±0.3 | 49.9±0.2 | **39.8**±0.7 | 66.5 |
| | | **Ours** | **47.7**±0.3 **+0.5** | **75.5**±0.4 **+1.5** | **51.8**±0.6 **+0.5** | **50.4**±0.3 **+0.5** | 39.7±0.9 **-0.1** | **68.0**±1.3 **+1.5** |
| Set 2 | 1 | FSRW [17] | 28.4±0.5 | 51.7±0.9 | 27.3±0.6 | 35.7±0.5 | 6.3±0.9 | 12.3 |
| | | FRCN+ft [39] | 30.3±0.5 | 49.7±0.5 | 32.3±0.7 | 38.8±0.6 | 5.0±0.6 | 9.4 |
| | | TFA [33] | 36.7±0.6 | 59.9±0.8 | 39.3±0.8 | 45.9±0.7 | 9.0±1.2 | 18.3 |
| | | DeFRCN [28] | 40.7±0.5 | 64.8±0.7 | 43.8±0.6 | 49.6±0.4 | 14.6±1.5 | 29.5 |
| | | **Ours** | **41.7**±0.9 **+1.0** | **66.8**±1.1 **+2.0** | **44.5**±1.1 **+0.7** | **50.5**±0.9 **+0.9** | **15.1**±2.3 **+0.5** | **31.8**±3.7 **+2.3** |
| | 2 | FSRW [17] | 29.4±0.3 | 53.1±0.6 | 28.5±0.4 | 35.8±0.4 | 9.9±0.7 | 19.6 |
| | | FRCN+ft [39] | 30.7±0.5 | 49.7±0.7 | 32.9±0.6 | 38.4±0.5 | 7.7±0.8 | 13.8 |
| | | TFA [33] | 39.0±0.4 | 63.0±0.5 | 42.1±0.6 | 47.3±0.4 | 14.1±0.9 | 27.5 |
| | | DeFRCN [28] | 42.7±0.3 | 67.7±0.5 | 45.7±0.5 | 50.3±0.2 | 20.5±1.0 | 39.7 |
| | | **Ours** | **43.6**±0.7 **+0.9** | **69.6**±0.9 **+1.9** | **46.6**±1.0 **+0.9** | **51.1**±0.4 **+0.8** | **21.2**±1.9 **+0.7** | **41.7**±2.5 **+2.0** |
| | 3 | FSRW [17] | 29.9±0.3 | 53.9±0.4 | 29.0±0.4 | 35.7±0.3 | 12.5±0.7 | 25.1 |
| | | FRCN+ft [39] | 31.1±0.3 | 50.1±0.5 | 33.2±0.5 | 38.1±0.4 | 9.8±0.9 | 17.4 |
| | | TFA [33] | 40.1±0.3 | 64.5±0.5 | 43.3±0.4 | 48.1±0.3 | 16.0±0.8 | 30.9 |
| | | DeFRCN [28] | 43.5±0.3 | 68.9±0.4 | 46.6±0.4 | 50.6±0.3 | 22.9±1.0 | 43.4 |
| | | **Ours** | **44.6**±0.6 **+1.1** | **70.9**±0.6 **+2.0** | **47.7**±0.7 **+1.1** | **51.4**±0.5 **+0.8** | **24.4**±1.2 **+1.5** | **46.6**±1.8 **+3.2** |
| | 5 | FSRW [17] | 30.4±0.4 | 54.6±0.5 | 29.5±0.5 | 35.3±0.3 | 15.7±0.8 | 31.4 |
| | | FRCN+ft [39] | 31.5±0.3 | 50.8±0.7 | 33.6±0.4 | 37.9±0.4 | 12.4±0.9 | 21.9 |
| | | TFA [33] | 40.9±0.4 | 65.7±0.5 | 44.1±0.5 | 48.6±0.4 | 17.8±0.8 | 34.1 |
| | | DeFRCN [28] | 44.6±0.3 | 70.2±0.5 | 47.8±0.4 | 51.0±0.2 | 25.8±0.9 | 48.1 |
| | | **Ours** | **45.2**±0.4 **+0.6** | **71.6**±0.5 **+1.4** | **48.3**±0.6 **+0.5** | **51.5**±0.4 **+0.5** | **26.4**±0.8 **+0.6** | **50.3**±1.3 **+2.2** |
| | 10 | FRCN+ft [33] | 32.2±0.3 | 52.3±0.4 | 34.1±0.4 | 37.2±0.3 | 17.0±0.8 | 39.7 |
| | | TFA [33] | 42.3±0.3 | 67.6±0.4 | 45.7±0.3 | 49.4±0.2 | 20.8±0.6 | 39.5 |
| | | DeFRCN [28] | 45.6±0.2 | 71.5±0.3 | 49.0±0.3 | 51.3±0.2 | **29.3**±0.7 | 52.8 |
| | | **Ours** | **45.9**±0.3 **+0.3** | **72.5**±0.3 **+1.0** | **49.1**±0.5 **+0.1** | **51.5**±0.2 **+0.2** | 29.1±0.8 **-0.2** | **53.7**±1.1 **+0.9** |
| Set 3 | 1 | FSRW [17] | 27.5±0.6 | 50.0±1.0 | 26.8±0.7 | 34.5±0.7 | 6.7±1.0 | 12.5 |
| | | FRCN+ft [39] | 30.8±0.6 | 49.8±0.8 | 32.9±0.8 | 39.6±0.8 | 4.5±0.7 | 8.1 |
| | | TFA [33] | 40.1±0.3 | 63.5±0.6 | 43.6±0.5 | 50.2±0.4 | 9.6±1.1 | 17.9 |
| | | DeFRCN [28] | 41.6±0.5 | 66.0±0.9 | 44.9±0.6 | 49.4±0.4 | 17.9±1.6 | 35.0 |
| | | **Ours** | **43.3**±1.0 **+1.7** | **69.1**±1.7 **+3.1** | **46.8**±1.1 **+1.9** | **50.9**±0.6 **+1.5** | **20.5**±3.7 **+2.6** | **39.6**±6.2 **+4.6** |
| | 2 | FSRW [17] | 28.7±0.4 | 51.8±0.7 | 28.1±0.5 | 34.5±0.4 | 11.3±0.7 | 21.3 |
| | | FRCN+ft [39] | 31.3±0.5 | 50.2±0.9 | 33.5±0.6 | 39.1±0.5 | 8.0±0.8 | 13.9 |
| | | TFA [33] | 41.8±0.4 | 65.6±0.6 | 45.3±0.4 | 50.7±0.3 | 15.1±1.3 | 27.2 |
| | | DeFRCN [28] | 44.0±0.4 | 69.5±0.7 | 47.7±0.5 | 50.2±0.2 | 26.0±1.3 | 38.3 |
| | | **Ours** | **45.3**±0.5 **+1.3** | **72.3**±0.6 **+2.5** | **48.6**±0.9 **+0.9** | **51.3**±0.4 **+1.1** | **27.6**±1.7 **+1.6** | **52.1**±2.4 **+13.8** |
| | 3 | FSRW [17] | 29.2±0.4 | 52.7±0.6 | 28.5±0.4 | 34.2±0.3 | 14.2±0.7 | 26.8 |
| | | FRCN+ft [39] | 32.1±0.5 | 51.3±0.8 | 34.3±0.6 | 39.1±0.5 | 11.1±0.9 | 19.0 |
| | | TFA [33] | 43.1±0.4 | 67.5±0.5 | 46.7±0.5 | **51.1**±0.3 | 18.9±1.1 | 34.3 |
| | | DeFRCN [28] | 45.1±0.3 | 70.9±0.5 | 48.8±0.4 | 50.5±0.2 | 29.2±1.0 | 52.9 |
| | | **Ours** | **46.2**±0.4 **+1.1** | **73.4**±0.5 **+2.5** | **49.4**±0.6 **+0.6** | **51.5**±0.3 **+1.0** | **30.5**±1.0 **+1.3** | **56.3**±1.9 **+3.4** |
| | 5 | FSRW [17] | 30.1±0.3 | 53.8±0.5 | 29.3±0.4 | 34.1±0.3 | 18.0±0.7 | 33.8 |
| | | FRCN+ft [39] | 32.4±0.5 | 51.7±0.8 | 34.4±0.6 | 38.5±0.5 | 14.0±0.9 | 23.9 |
| | | TFA [33] | 44.1±0.3 | 69.1±0.4 | 47.8±0.4 | 51.3±0.2 | 22.8±0.9 | 40.8 |
| | | DeFRCN [28] | 46.2±0.3 | 72.4±0.4 | 50.0±0.5 | 51.0±0.2 | 32.3±0.9 | 57.7 |
| | | **Ours** | **47.2**±0.4 **+1.0** | **74.5**±0.5 **+2.1** | **50.8**±0.6 **+0.8** | **51.8**±0.3 **+0.8** | **33.5**±0.9 **+1.2** | **60.3**±1.2 **+2.6** |
| | 10 | FRCN+ft [39] | 33.1±0.5 | 53.1±0.7 | 35.2±0.5 | 38.0±0.5 | 18.4±0.8 | 44.6 |
| | | TFA [33] | 45.0±0.3 | 70.3±0.4 | 48.9±0.4 | 51.6±0.2 | 25.4±0.7 | 45.6 |
| | | DeFRCN [28] | 47.0±0.3 | 73.3±0.3 | 51.0±0.4 | 51.3±0.2 | 34.7±0.7 | 60.8 |
| | | **Ours** | **47.8**±0.3 **+0.8** | **75.1**±0.3 **+1.8** | **51.6**±0.5 **+0.6** | **51.9**±0.2 **+0.6** | **35.6**±1.2 **+0.9** | **63.3**±1.2 **+2.5** |

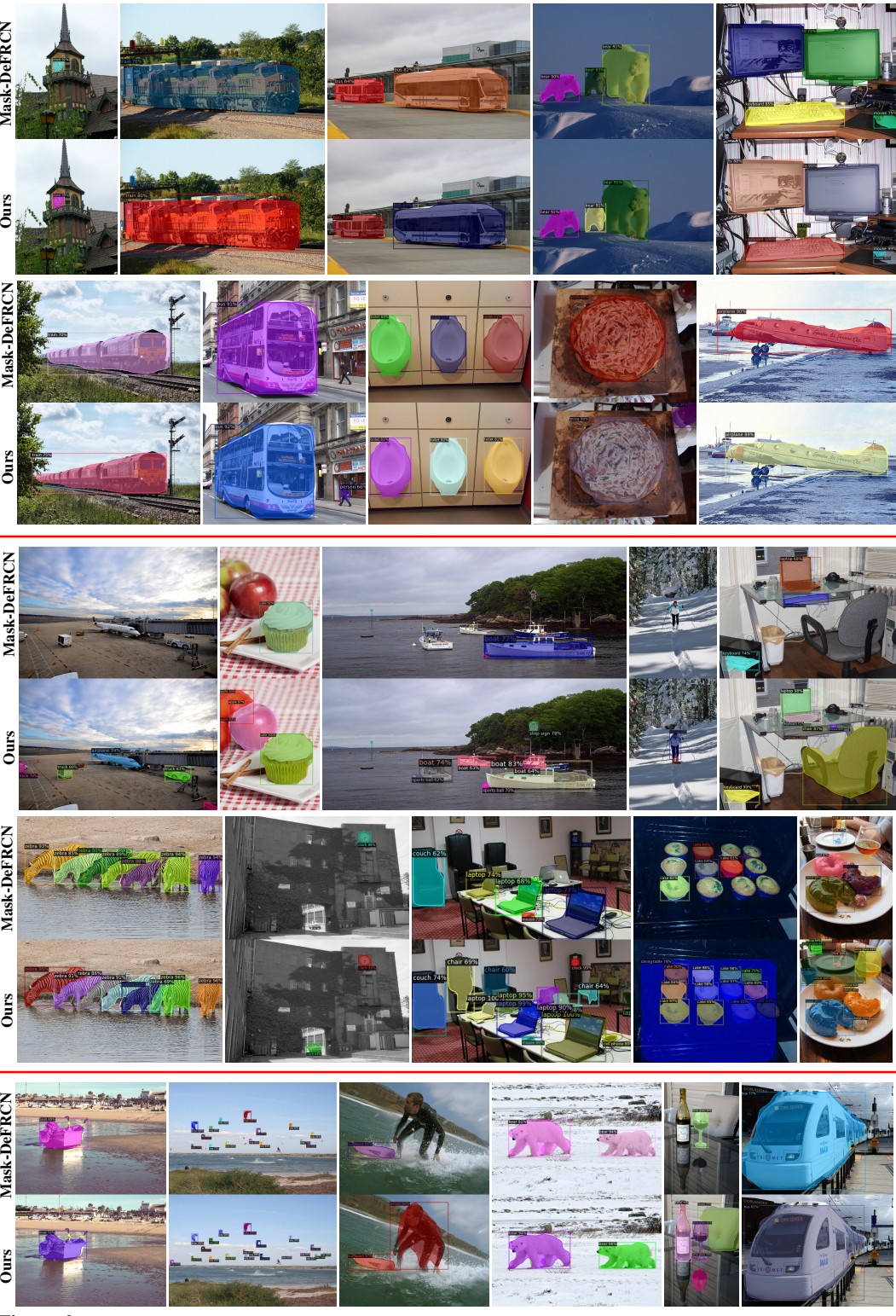

Figure 6: Visualization results of our method and the strong baseline (Mask-DeFRCN) on MS-COCO validation images under the gFSIS setting with $K$=10. These bounding boxes and segmentation masks are visualized using classification scores larger than 0.6. The top two rows show success cases with our method and the baseline while the middle two rows show success cases with our method but partly failure ones with the baseline. The baseline may tend to incorrectly recognize positive object regions as background due to the biased classification. The bottom row shows some failure cases from left to right, small objects (e.g., the small boats and the person), coarse boundary segmentation (e.g., the surfer), occlusion (e.g., two bears are detected to one), and misclassification of similar appearance objects (e.g., the shadow of wine glass is recognized to wine glass and the train is detected to bus). Best viewed in color and zoom in.

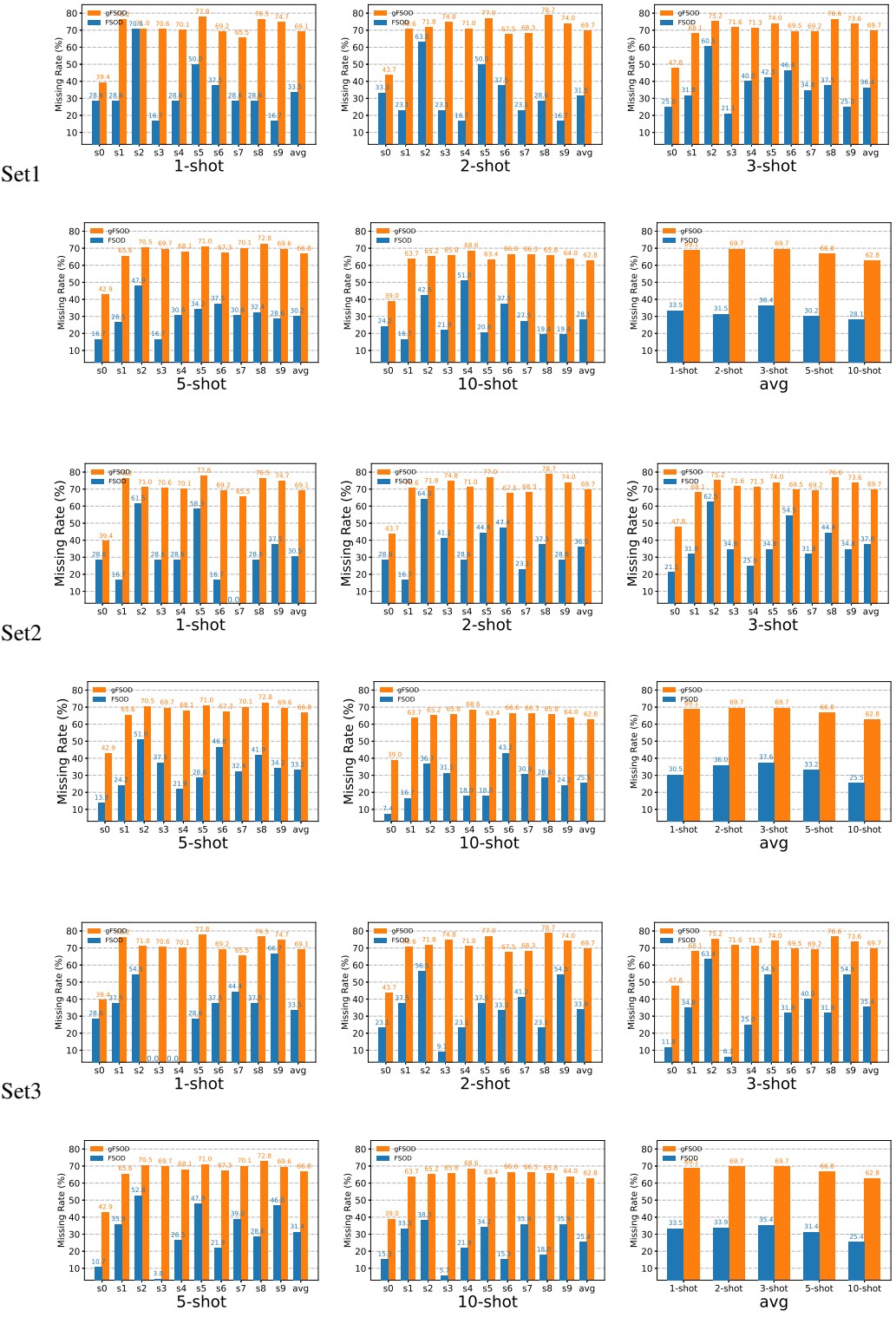

Figure 7: Comparisons of the proportion of missing labeled instances of FSOD and gFSOD on the PASCAL VOC dataset. Although PASCAL VOC is simpler than MS-COCO, there are still similar observations (high missing rates) on the PASCAL VOC dataset. Different from the MS-COCO, the missing rate is the same among three sets on each shot for the gFSOD setting.