# OpenReview forum: "Decoupling Classifier for Boosting Few-shot Object Detection and Instance Segmentation"
_NeurIPS.cc/2022/Conference — NeurIPS 2022 Accept_

### Official Review · Reviewer_DeAx · 2022-07-08

**Rating:** 7
**Confidence:** 5
**Soundness:** 3 good
**Presentation:** 3 good
**Contribution:** 3 good

**Summary:**

The paper formally proposes the missing label issue which naturally happens in few-shot scenarios including few-shot object detection (FSOD) and few-shot instance segmentation (FSIS). The missing label issue may result in biased classification (mistakenly recognizing novel class objects as background, i.e., missing detection) and thus reduce the generalization performance of many current FSOD/FSIS models.  From the missing label perspective, the paper presents a simple but effective method that decouples the standard classifier into two parallel heads which are capable of addressing clear positive samples and noisy negative samples, respectively. Using the proposed decoupling classifier, the model can effectively learn novel classes because the effects of noisy negative samples are well mitigated. The effectiveness of the proposed method is evaluated on standard few-shot benchmarks PASCAL VOC and MS COCO for both FSOD/FSIS and generalized FSOD/FSIS tasks.

**Questions:**

- This missing label issue should be comprehensively analyzed from a broad perspective, e.g., between base and novel classes stages, the novel fine-tuning stage itself.

- The inference pipeline (using the standard classifier or the decoupling classifier) should be more clarified.


**Limitations:**

Yes. The authors have discussed the limitations of the proposed methods in Sec.4 (Experiments) and Sec.5 (Conclusion). As stated in the paper, It may not be suitable when the missing label rate is small. However, the proposed method is still comparable to its counterpart even if the missing label rate is zero, which indicates the robustness of the proposed method. Considering that it is unavoidable to meet the missing label issue in few-shot FSOD and FSIS, especially under multiple categories scenarios and so it is ok for me.

**Strengths And Weaknesses:**

Strengths:
+ The paper observes the fact that the missing label issue naturally exists in few-shot scenarios such as FSOD and FSIS.  Furthermore, the authors analyze the missing label issue may result in biased classification and reduce the generalization performance of FSOD and FSIS. This new perspective might be inspiring to others in the field.

+ In order to mitigate the effects of noisy negative samples and reduce the biased classification in FSOD and FSIS, the paper proposes a novel classifier decoupling idea that is quite interesting and strongly motivated. It can individually address clear positive samples and noisy negative samples and thus mitigate the biased classification issue. The proposed method is really simple and easy to understand and follow.

+ The performance of the proposed method outperforms its baseline and state-of-the-art by a large margin on PASCAL VOC and MS-COCO benchmarks for both FSOD/FSIS and generalized FSOD/FSIS tasks. Meanwhile, the proposed method doesn’t introduce any additional computation costs and hyper-parameters because it only simply decouples the standard classifier into two parallel classifiers. Therefore, it seems like a practical algorithm that can be put immediately into practice.

+ The proposed method is well analyzed and interpreted from the gradient optimization perspective. On the other hand, the authors evaluate the performance of the learned classifier using the Recall metric for all ground-truth foreground objects of all testing images and demonstrate that the proposed decoupling classifier indeed mitigates the biased classification.

+ Ablation studies cover all the crucial components of the proposed approach. The experiment results are quite convincing.

+ The manuscript is well written and the organization is very clarified.

Weaknesses:
- Relationship with some existing works~(e.g., DeFRCN). The missing label issue also, in fact, exists between base and novel classes. For example, the novel class objects may potentially present in base images at the base learning stage but these potentially novel objects are viewed as background and it is opposite to the base learning stage at the novel fine-tuning stage. Therefore, the missing label issue also leads to foreground-background confusion between base and novel learning stages because of this two-stage fine-tuning mechanism. I understand that this paper focuses on the missing label issue at the novel fine-tuning stage. However, it is more helpful for the readers to understand the proposed method if these two types of missing label issues can be comprehensively analyzed.

- It is unclear how to use the decoupling classifier at the inference stage because the image-level few-shot label ($m_i$ in Eq. 7) is agnostic for any testing images. I guess that the standard classifier is used at the inference stage and the proposed decoupling classifier only is employed during training time. The authors should clarify it.

---

> ### Author Response · Authors · 2022-08-02
> **Response to Reviewer DeAx**
>
> We sincerely thank you for your detailed comments and constructive suggestions, especially your appreciation for our work, "**this new perspective might be inspiring to others in the field**", "**novel classifier decoupling idea that is quite interesting and strongly motivated**", "**really simple and easy to understand and follow**", "**outperforms its baseline and state-of-the-art by a large margin**", "**well analyzed and interpreted**", "**experiment results are quite convincing**" and "**well written and very clarified**". Next, we respond to your concerns as follows.
>
> **Q1:This missing label issue should be comprehensively analyzed from a broad perspective, e.g., between base and novel classes stages, the novel fine-tuning stage itself.**
>
> Thanks for your suggestions. We agree that DeFRCN also can be interpreted from a missing label perspective between base and novel classes. Based on this, we could view fine-tuning few-shot learning paradigm as a domain adaption procedure from base to novel. In this procedure, few-shot detector may suffer from foreground-background confusion because one proposal (potential novel object) belongs to background (negative class) in the base learning stage and becomes foreground (positive class) in the novel fine-tuning phase. To mitigate the label conflict between two domains, DeFRCN decouples RCNN and RPN by stopping gradient backpropagation of RPN in Faster-RCNN. Different from the missing label of cross-domain in DeFRCN, we focus on the missing label issue in the novel (or balanced base-novel) fine-tuning stage. We will comprehensively analyze the missing label issue from a broad perspective and add this disscussion to the next version.
>
> **Q2:The inference pipeline (using the standard classifier or the decoupling classifier) should be more clarified.**
>
> Thanks for your suggestions. Your analysis is correct. At inference time, we only use the positive head as the same as the standard classifier for all samples, while the proposed decoupling classifier is used only during training. We add it to the next version for clarification.
>
> We thank you again for your time and efforts in reviewing our paper. Furthermore, we would be more than happy to discuss with you if you have any concerns about our responses.

---

> > ### Comment · Reviewer_DeAx · 2022-08-07
> > **Thanks for your reply.**
> >
> > I appreciate the authors for their responses to my concerns. This paper contributes a simple but effective method (decoupling classifier) for few-shot object detection and few-shot instance segmentation from a new missing label perspective, which is also approved by other reviewers. The main concern is the training and evaluation protocol (benchmarks) in few-shot detection and instance segmentation, which is well clarified in the rebuttal after carefully reading the comments of other reviewers and the feedback of the authors.
> >
> > In summary, I will raise my score from weak accept to accept, in expectation of your inclusion of some discussion of the missing label issue from a broad perspective in the final version.

---

> > > ### Author Response · Authors · 2022-08-07
> > > **Thasnks Reviewer DeAx for your strong support  for our work**
> > >
> > > We greatly appreciate the strong support for our work (increasing your score) and the positive comments for contributing a simple but effective FSOD and FSIS method from a new missing label perspective. We will certainly include discussions of the missing label issue from a broad perspective in the updated version of our paper.

---

### Official Review · Reviewer_mATA · 2022-07-10

**Rating:** 3
**Confidence:** 4
**Soundness:** 2 fair
**Presentation:** 2 fair
**Contribution:** 2 fair

**Summary:**

This paper proposes a decoupled classifier to deal with positive and negative proposals separately when learning general few-shot object detection (FSOD) and few-shot instance segmentation (FSIS) networks. Specifically, the paper argues that for FSOD/FSIS problem, missing labels are a common issue. For an image, some classes could be ignored during annotation. The missing annotation problem would lead to a biased classifier, especially when learning with a few images. To solve this problem, the paper proposes to deal with positive proposals with normal cross-entropy loss and deal with negative proposals by only considering the positive class and background of the images.

The proposed method is implemented based on the architecture of DeFRCN [21] and evaluated on benchmark datasets VOC 15/5 and COCO 60/20. The method is compared with several SOTA methods, including DeFRCN, and appears to be better.

Overall, the idea of the paper should be easily reproducible. The benchmark, evaluation, and comparison are solid. However, I think there are several major issues with the premise, and the proposed method could lead to more bias in certain situations. Details can be found in the  Strengths And Weaknesses section.

**Questions:**

Please refer to the weakness section for questions.

**Strengths And Weaknesses:**

The main strengths of the paper are:

1. The paper is relatively easy to follow. The proposed method is described in detail and should be easily reproducible.
2. The comparisons and evaluation are adequate, and the results are positive on the benchmark.

The main weaknesses of the paper are:

1. My major concern with the paper is the premise. I think the paper proposes a solution to deal with the bias of existing benchmark datasets rather than the problem (FSOD/FSIS) themselves. In particular, in Fig 1(c), the paper mentions that for a one-shot image, 'dog' could be annotated and the 'person' could be ignored.  This situation is mainly due to how existing benchmarks (VOC/COCO) are proposed. Prior works [14, 26] proposed the benchmarks that consider an object instance (bounding boxes) as a 'shot'. In this case, the training images could just be sampled with one instance, even though there are multiple instances in the same image. I agree with the paper that this is an issue with these benchmarks and needs to be addressed. However, I do not think this is an inherent issue with FSOD/FSIS problems.
2. Even if the problem is inherent with FSOD, the proposed solution could create more bias rather than solving the problem. With the same example in Fig 1(c). If it is a one-shot scenario for the class 'person', then the other person might not be annotated. In fact, this is a very common problem for object detection, even for large-scale datasets. There are plenty of examples of 'group' bounding boxes with a single example instance in the OpenImages dataset (https://storage.googleapis.com/openimages/web/index.html).  These missing annotations would be considered negative ('background') and the proposed method restricted the learning to between ['person', 'background'], and backpropagate wrong losses. I think in many situations, the proposed method can lead to more bias and confusion.
3. Some of the paper's claims are not well-grounded. For example, Line 94-95, the paper claims that " It is impractical from the perspective of many real-world applications because ones always expect that few shot model is capable of not only recognizing novel classes but also remembering base classes". However, for many few-shot learning papers (MAML [a] and Prototypical Network [b] for example), base-class performance is not a concern. Could the author motivate more on why we 'always expect the few-shot model to remember base classes‘？

[a] Model-Agnostic Meta-Learning for Fast Adaptation of Deep Networks, Finn et al, ICML 2017
[b] Prototypical Networks for Few-shot Learning,  Snell at al, NeurIPS 2017

---

> ### Author Response · Authors · 2022-08-02
> **Response to Reviewer mATA. (Part 2/2)**
>
> **Q2: The proposed solution could create more bias rather than solving the problem.**
>
> We are sorry for resulting in this misunderstanding. Here, we try to explain this issue as follows.
>
> First, the proposed decoupling classifier is simple but effective, it significantly boosts FSOD and FSIS performance (Tables 1, 2, 3, 4, 5 and 6). Meanwhile, we also analyze why it works from two perspectives including gradient optimization (Sec 3.3 and Fig. 2) and the generalization ability of the learned classifier (Sec 4.3 and Fig. 3). These above points are also recognized by you ("**the idea is easily reproducible**", "**evaluation, and comparison are solid**") and other reviewers ("**simple yet effective**", "**SOTA results**", "**solid quantitative evidence**", "**a plugin for many FSOD methods**" and "**well analyzed and interpreted**").
>
> Second, it is unreasonable and unfair to judge our method which could create more bias (no evidence). In contrast, our proposed method significantly mitigates the biased classification when meeting the missing label FSOD and FSIS (Sec 4.3 and Fig. 3). What is more, our method still achieves comparable performance even if the missing label rate is small or zero (bottom two rows in Table 5), which indicates its robustness. We have discussed it (lines 293-300) and taken it as a limitation of our method (lines 346-348).
>
> Third, it is important to work for our method depending on a premise, i.e., missing label few-shot scenarios. And it is really unnecessary to use our method in a fully-annotated scenario no matter large-scale or few-shot. Here we give some explanations for clarification.
> - As shown in Fig 1(c), only the “dog” instance is labeled and the other two "person" instances are missing labeled. Once one missing labeled “person” instance is sampled and it will be mistakenly taken as a background class in a standard classifier. Using the proposed decoupling classifier, the person instance will be fed to a negative head and its optimization is restricted between “dog” and “background” (Eq. 13). Therefore, the biased classification is mitigated.
> - For a special case as mentioned by you, for instance, multiple instances present in an image and belong to the same class, e.g. person class, and only one person instance is annotated. We agree with you that it is difficult for our decoupling classifier to correct the learning for this type of missing instance. It is worth noting that the standard classifier also can't solve this type of case. Our decoupling classifier only degenerates into the standard classifier when meeting this special case, but it doesn't imply that our method creates bias. This case may be present in the current benchmarks because of random sampling mechanism. However, the overall performance has been significantly improved by using our method.
>
> In summary, our decoupling classifier doesn't involve the wrong backpropagation compared to the standard classifier. In contrast, the negative head of our decoupling classifier corrects or mitigates the wrong backpropagation. This is also why the decoupling classifier works.
>
> **Q3: Some of the paper's claims are not well-grounded. For many few-shot learning papers (MAML and Prototypical Network), base-class performance is not a concern. Could the author motivate more on why we 'always expect the few-shot model to remember base classes‘？**
>
> We agree with you that many previous few-shot learning methods focus on the performance of novel classes only. But, recent **generalized** few-shot learning considers the performance **not only on novel classes but also on base classes** [1,2].  The papers [1,3] stress that a good few-shot learning system should adapt to new tasks rapidly while maintaining the performance on previous knowledge without forgetting. Recently, some state-of-the-art methods (e.g., TFA and DeFRCN) evaluate their performance not only with FSOD setting but also with generalized FSOD (gFSOD). We follow these works and report performance for FSOD and FSIS under these two settings (FSOD/FSIS and gFSOD/gFSIS).
>
> [1] Dynamic Few-Shot Visual Learning without Forgetting, CVPR 2018.
>
> [2] Generalized Few-Shot Object Detection without Forgetting, CVPR 2021.
>
> [3] Gradient episodic memory for continual learning, NeurIPS 2017.
>
> We thank you again for your time and efforts in reviewing our paper. Based on the above clarifications, we sincerely hope from the heart that you could re-evaluate our work. Furthermore, we would be more than happy to discuss with you if you have any concerns about our responses.

---

> ### Author Response · Authors · 2022-08-02
> **Response to Reviewer mATA. (Part 1/2)**
>
> We sincerely thank you for your detailed comments and positive feedback on our paper, “**the idea of the paper is easily reproducible**” and “**the benchmark, evaluation, and comparison are solid**”. Next, we respond to your concerns in a point-by-point manner as follows.
>
> **Q1: I agree with the paper that this is an issue with these benchmarks (the standard evaluation protocol in FSOD/FSIS) and needs to be addressed. However, I do not think this (missing label) is an inherent issue with FSOD/FSIS problems.**
>
> Thanks for your endorsement of the FSOD/FSIS benchmarks (training and evaluation protocol) used in our paper. Here, we want to add some explanations below and hope to address your concerns about the missing label issue in FSOD/FSIS.
>
> First, we follow state-of-the-art FSOD/FSIS methods and use the standard benchmarks which have been widely accepted and used in the community of machine learning and computer vision for a fair comparison. To the best of our knowledge, recent published FSOD/FSIS papers almost use the same benchmarks that consider an object instance as a “shot”. As pointed out by you,  there are generally multiple instances in an image for instance-level FSOD and FSIS, which is different from image-level few-shot image classification.
>
> Second,  the community accepts the current FSOD/FSIS benchmarks because they are more challenging due to incomplete or partial annotations (i.e., missing labels), and the number of labeled instances is well controlled in each class for fair comparison of few-shot instance-level recognition methods. The missing label issue requires that learning algorithms deal with training images each associated with multiple instances, among which only partial instances are labeled, which is also similar to partial label learning. As we know, missing label (partial label) learning is more difficult and challenging than conventional fully-supervised learning, especially few-shot scenarios.
>
> Last but at least, we think that it is generally expensive and time-consuming to label all instances in many real-world applications. Fully-supervised object detection or instance segmentation typically assumes that all interest instances are labeled for given training images. In some real-world applications, e.g., open-vocabulary object detection, however, it is generally challenging to label all instances, and thus there still exists some instances left to be missing labeled, although we agree that it is possible to label all instances given few-shot images. In addition, it is more friendly and convenient for users to label partial instances than all instances even in few-shot scenarios.

---

> ### Author Response · Authors · 2022-08-07
> **further discussion**
>
> Dear reviewer mATA,
>
> Thank you for taking the time to review our work. We hope our rebuttal has addressed your questions and concerns. We would be more than happy to discuss with you if you still have any unresolved concerns or additional questions about the paper or our rebuttal.
>
> Best,

---

### Official Review · Reviewer_1weo · 2022-07-12

**Rating:** 4
**Confidence:** 4
**Soundness:** 3 good
**Presentation:** 3 good
**Contribution:** 2 fair

**Summary:**

This paper addresses the missing label issue in few-shot object detection (FSOD) and few-shot instance segmentation (FSIS). Some foreground examples are not labeled in the few-shot scenario, which causes classification bias for the conventional classification head used in object detectors. This paper proposes a simple yet effective algorithm that treats the foreground and background proposals differently in the classification loss. The proposed method is evaluated on benchmark datasets and shows competitive performance against SOTA methods for both FSOD and FSIS.

**Questions:**

1. If the novel class training samples are provided in a way that every example is labeled in each image but only a small group of images are labeled (basically the strategy I mentioned in Cons), do you think the proposed method works better or worse? Also, will other FSOD/FSIS methods work better or worse in this case? Do you think using the proposed method with the current labeling strategy (i.e. randomly labeling a single class and ignoring others in one image) is still more beneficial as it sees more data?

**Limitations:**

The authors discussed the limitations

**Strengths And Weaknesses:**

Pros:

1. The proposed method is very simple yet effective. It decouples the loss function for foreground and background objects to remedy the effect on the classification head from the missing labels for potential foreground objects. The evaluations on benchmark datasets demonstrate its effectiveness in both FSOD and FSIS.
2. The authors show that the proposed loss can be incorporated into other FSOD methods to achieve better performance. It also has no negative impact on base classes per the evaluation results. It might become a plug-in for many FSOD methods.
3. The paper is generally well written and easy to follow.

Cons:

1. While I acknowledge the issue of missing labels in the FSOD, it is more like a bad design in the current FSOD evaluation protocol rather than a real-world problem. It is weird when you have a lot of images, but you only label a single class and ignore others, especially when you know it will have a negative impact. If you have limited resources (as in a few-shot setting), shouldn't you focus on labeling a small group of images to ensure every class is labeled so that the total # labels are roughly similar? I understand the protocol is designed to control the number of shots, but it just seems unrealistic.

---

> ### Author Response · Authors · 2022-08-01
> **Response to Reviewer 1weo**
>
> We sincerely thank you for your appreciation of our work “**very simple yet effective**”, “**achieving better performance and no negative impact on base classes**”, “**become a plug-in for many FSOD methods**” and “**well written and easy to follow**”. Next, we respond to your concerns in a point-by-point manner as follows.
>
> **Q1: If the novel class training samples are provided in a way that every example is labeled in each image but only a small group of images are labeled (basically the strategy I mentioned in Cons), do you think the proposed method works better or worse?**
>
> Thanks for your suggested setting that all instances are fully annotated for given few-shot training images.
>
> The suggested setting means the missing label rate is zero. In fact, our method still achieves comparable performance even if the missing label rate is small or zero (bottom two rows in Table 5), which indicates the robustness of our method. The detailed discussion can be found in Sec. 4.2: few-shot object detection on the PASCAL VOC and MS-COCO (lines 293-300). And we have discussed it and taken it as a limitation of our method (lines 346-348).
>
> In this paper, our work mainly focuses on the missing label few-shot scenario (current and popular benchmark) and achieves significant improvements under two different settings (FSOD/FSID and gFSOD/gFSIS). We leave fully annotated (building new benchmarks) FSOD and FSIS as our future work.
>
> **Q2: Also, will other FSOD/FSIS methods work better or worse in this case?**
>
> We try to take the TFA as our baseline and replace the standard classifier in TFA with the proposed decoupling classifier. And we find that the performance improvements are consistent with that of DeFRCN. The detailed discussions refer to our response to Q6 of reviewer LjLj.
>
> **Q3: Do you think using the proposed method with the current labeling strategy (i.e. randomly labeling a single class and ignoring others in one image) is still more beneficial as it sees more data?**
>
> Yes, it is still beneficial when using the randomly sampling mechanism (randomly labeling a single class and ignoring others in one image).
>
> In this paper, we follow standard FSOD/FSIS training and evaluation protocol which have been widely used in the community of machine learning and computer vision. In the standard protocol, the labeled instances are generated, in fact, by randomly sampling. The number of sampled instances is decided by giving shot numbers. For example, 1-shot implies that only one instance is sampled in giving an image and other instances will be ignored in this image. We have reported 1-shot results based on multiple random seeds for PASCAL VOC and MS-COCO in Tables 1, 2, 3, 4, 5 and 6.  It can be seen that the improvements in our method are consistent with a 1-shot setting.
>
> We thank you again for your time and efforts in reviewing our paper. Furthermore, we would be more than happy to discuss with you if you have any concerns about our responses.

---

> ### Author Response · Authors · 2022-08-07
> **further discussion**
>
> Dear reviewer 1weo,
>
> Thank you for taking the time to review our work. We believe that our rebuttal has addressed your questions and concerns. We would be more than happy to discuss with you if you have any further questions about the paper or our rebuttal.
>
> Best,

---

### Official Review · Reviewer_LjLJ · 2022-07-15

**Rating:** 7
**Confidence:** 3
**Soundness:** 3 good
**Presentation:** 3 good
**Contribution:** 3 good

**Summary:**

The paper proposes a method of correcting the missing label issue in the context of few-shot instance segmentation and few-shot object detection problems.

**Questions:**

- The notation in equations 11-13 is confusing: equation 11 is the derivative of $L_{cls}^{fg}$, equation 12 is the derivative of $L_{cls}^{fg}$ and equation 13 is the derivative of $L_{cls}^{bg}$, while on the LHS there is always $L_{cls}$
- How easy it is to extend other existing backbones with the proposed technique? What does it take in terms of actual code changes? Can you provide a clean code example?

**Ethics Review Area:**

["I don’t know"]

**Strengths And Weaknesses:**

Strengths:

- The paper is very well focused on solving one specific issue
- reasonably clearly written
- empirical framework is solid
- SOTA results
- Section "Why DC works?" seems to provide solid quantitative evidence to support the main claim of the paper.

Weaknesses:

- Contribution is rather shallow and incremental, the paper almost reads as a workshop paper in some respects. It is hard to evaluate the significance of the work, because problem being solved is not well articulated (see continuation below)
- problem being solved is not well articulated
  - In Fig 1 it is unclear why one of the person instances is mislabeled
  - it is unclear if the label noise is due to the dataset deficiency or it is due to the incorrect sampling scheme used to train few-shot methods. Is it possible to rectify the sampling scheme for few shot to avoid label noise? How does it happen that the objects that are not supposed to used as supervision signals end up being used as such?
  - It is unclear why this problem is solved architecturally by introducing additional heads while it seems like it is really a problem with the data. I would much prefer if the problem could be solved in the source, while in this case it looks like the data problem remains and instead of solving it directly, a heuristic is added at the model level to account for it
- Missing qualitative examples showing when the proposed technique succeeds in solving the stated problem and when it fails
- Since the proposed technique is claimed to solve a high level issue, applying it to a few base methods and showing that it works seems very important. However, this paper only shows results based on Mask-DeFRCN.

---

> ### Author Response · Authors · 2022-08-01
> **Response to Reviewer LjLJ (Part 3/3)**
>
> **Q6: This paper only shows results based on Mask-DeFRCN.**
>
> We take DeFRCN as our baseline and extend it from FSOD to FSIS in our initial submission. One main consideration is its simple framework and state-of-the-art performance for FSOD. Here we report the performance that replaces the baseline DeFRCN with TFA on MS-COCO with 1-, 5-, and 10-shot. Specifically, we only replace the standard classifier in TFA with the proposed decoupling classifier and keep other components and hyper-parameters unchanged. As shown in the Table below, our re-produced results (denoted as TFA*) are close to that of the original TFA paper both on novel and base classes. What is more, the proposed decoupling classifier helps to improve the baseline TFA by about 1 point in AP (2-3 points in AP50) on novel classes and 3-4 points in AP (5-6 points in AP50) on base classes across different shots settings. These results are consistent with that of plugging our decoupling classifier into DeFRCN.
>
> | Shot | 1  | 5 | 10 |
> |---|:---:|:---:|:---:|
> | **Novel**|  AP AP50  | AP AP50 | AP AP50 |
> | TFA (reported in paper) | 1.9 3.8 | 7.0 13.3| 9.1 17.1 |
> | TFA* (re-produced by us) | 1.6 3.5 | 6.8 13.3| 9.0 17.2 |
> | **Ours**      | **2.5** **5.6** | **7.8** **15.9**| **10.0** **20.0** |
>
> | Shot |  1  |  5 | 10 |
> |---|:---:|:---:|:---:|
> | **Base** |  AP AP50  | AP AP50 | AP AP50 |
> | TFA  (reported in paper)        |31.9 51.8 | 32.3 50.5| 32.4 50.6 |
> | TFA*  (re-produced by us) |31.9 51.3 | 31.7 49.7| 31.9 49.9 |
> | **Ours**       |**34.6** **56.9** | **35.2** **56.0**| **35.0** **55.6** |
>
>
> **Q7: The notation in equations 11-13 is confusing: equation 11 is the derivative of Lclsfg, equation 12 is the derivative of Lclsfg and equation 13 is the derivative of Lclsbg, while on the LHS there is always Lcls.**
>
> We thank you for pointing out the typos in Eq. 12 and 13. Note that the $L_{cls}$ consists of two heads, $L_{cls}^{fg}$ and $L_{cls}^{bg}$ in Eq. 6. Therefore, the derivate of $L_{cls}$ with respect to $\theta_{cls}$ will also consists of two parts, i.e., Eq. 12 and 13. Note that the name of $L_{cls}$ may be misleading in Eq. 12 and 13, thus we will modify it as $L_{cls}^{fg}$ in Eq. 12 and $L_{cls}^{bg}$ in Eq. 13 in next version.
>
> **Q8: How easy it is to extend other existing backbones with the proposed technique? What does it take in terms of actual code changes? Can you provide a clean code example?**
>
> It is easy to use other backbones with the proposed method because we only simply modify the classifier of DeFRCN (Faster-RCNN framework) which originally supports various backbones. Considering state-of-the-art DeFRCN reports results based on the ResNet-101 backbone, we follow it and also report experimental results based on the same backbone for fair comparisons.
>
> Thank you very much for your attention to the actual code implementation. As pointed out by other reviewers, our method is very simple and easy to understand/follow/reproduce. In addition, we also promise that the code will be available (line 15) in our submission. Here, we provide a PyTorch style code for the proposed decoupling classifier as follows.
> ```python
> def dc_loss(x, y, m):
>     """
>     Compute the decoupling classifier loss.
>     Return scalar Tensor for a single image.
>
>     Args:
>         x: predicted class scores in [-inf, +inf]，x’s size: (N, C+1), (N is the number of region proposals of each image)
>         y: ground-truth classification labels in [1, C+1], where [1, C] represent foreground object classes and C+1 represents the background class, y’s size (N,1)
>         m: image-level label vector and its element is 0 or 1, m’s size: (1, C+1)
>
>     Returns：
>         loss
>     """
>     # background class index
>     N = x.shape[0]
>     bg_label = x.shape[1]-1
>
>     # positive head
>     pos_ind = y!=bg_label
>     pos_logit = x[pos_ind,:]
>     pos_score = F.softmax(pos_logit, dim=1) # Eq. 4
>     pos_loss = F.nll_loss(pos_score, y[pos_ind], reduction="sum") #Eq. 5
>
>     # negative head
>     neg_ind = y==bg_label
>     neg_logit = x[neg_ind,:]
>     neg_score = F.softmax(m.expand_as(neg_logit)*neg_logit, dim=1) #Eq. 8
>     neg_loss = F.nll_loss(neg_score, y[neg_ind], reduction="sum")  #Eq. 9
>
>     # total loss
>     loss = (pos_loss + neg_loss)/N
>     return loss
> ```
> It can be seen that the main change is to only introduce an image-level label vector ($\vec m$）on the standard softmax function for the negative head and others keep unchanged like the positive head, but the performance improvements are consistent.
>
> We thank you again for your time and efforts in reviewing our paper. Furthermore, we would be more than happy to discuss with you if you have any concerns about our responses.

---

> ### Author Response · Authors · 2022-08-01
> **Response to Reviewer LjLJ (Part 2/3)**
>
> **Q3: Is it possible to avoid label noise? How does it happen that the objects that are not supposed to used as supervision signals end up being used as such?**
>
> Yes, it is possible to avoid label noise in a few-shot setting. For example, the community could build new benchmarks that all interest instances are fully annotated in given few-shot training images. However, designing a new benchmark is beyond our current research because we mainly focus on few-shot with missing label scenarios. Furthermore, we argue that current benchmarks are more challenging than the fully-annotated ones. The reason is that, as we know, missing label (partial label) learning is more difficult and challenging than conventional fully-supervised learning, especially in few-shot settings. In addition, we want to emphasize that it is often expensive and time-consuming to completely annotate all interest instances in practical applications. So it is more friendly and convenient for users to utilize the miss-labeled protocol (the current benchmarks).
>
> The missing label (noisy) instances may bring biased classification and finally result in missing detection. And we have some corresponding discussions and analyses in Sec. 3.2: Biased classification issue (lines 146-163). Here we provide more explanations. In a two-stage object detection framework, e.g., Faster RCNN, positive (foreground) and negative (background) samples are generated by computing IoU scores between all RPN proposals and ground-truth bounding boxes. Under this strategy, those missing labeled instances will be mistakenly assigned as negative labels, but they are truly positive. Therefore, the model will be misguided by these noisy negative samples during training and biased towards background at inference time. This potentially limits the generalization ability of the few-shot model on novel classes.
>
> **Q4: It is unclear why this problem is solved architecturally by introducing additional heads while it seems like it is really a problem with the data. I would much prefer if the problem could be solved in the source.**
>
> Our method decouples the standard classifier into two parallel heads (positive and negative) to independently process clear positive samples and noisy negative samples for mitigating the biased classification for FSOD/FSIS and gFSOD/gFSIS. We analyze why it works from two aspects, gradient optimization (Sec 3.3) and the generalization ability of the learned classifier for foreground objects (Sec 4.3). We understand that it will not involve the missing label issue if all instances are fully annotated. Unfortunately, this requires a well-annotated dataset which leads to a chicken-and-egg problem–we need a fully-annotated dataset to train a good few-shot model, but we need a good few-shot model with a partly-annotated dataset.
>
> **Q5: Missing qualitative examples.**
>
> Thanks for your suggestion. We show qualitative results including success and failure cases and compare our method with its baseline in the supplementary material (Fig. 1) due to the page space limitation. As shown in Fig. 1, we can observe that the baseline method may tend to incorrectly recognize foreground objects as background due to the biased classification (middle part in Fig. 1). In addition, our method also may produce failure results due to small objects, occlusion, and misclassification of similar appearance objects (bottom part in Fig. 1).

---

> ### Author Response · Authors · 2022-08-01
> **Response to Reviewer LjLJ (Part 1/3)**
>
> We sincerely thank you for the detailed comments and positive feedback such as “**solid framework**”, “**SOTA results**”, “**solid quantitative evidence to support the main claim**”, and "**clear writing**". For each detailed question, we provide responses below.
>
> **Q1: Contribution is rather shallow and incremental.**
>
> The main contribution of our paper has been well summarized and recognized by other reviewers (1weo and DeAx), and here we want to emphasize them again as follows:
>
> Firstly, we rethink FSOD and FSIS tasks and discover that the existing fine-tuning-based few-shot FSOD and FSIS methods severely suffer from biased classification because of the observed missing label issue. To the best of our knowledge, this is **the first time to propose the missing label issue in FSOD and FSIS from a label completeness perspective**. **This new perspective might be inspiring to others in this field** as pointed out by reviewer DeAx. For example, benchmark itself (commented by other reviewers) in FSOD and FSIS.
>
> Secondly, we propose a **very simple** (core implementation **only with one line code**, Eq. 8) **but effective** (e.g., **5.6+** AP50 improvements for detection and **4.5+** AP50 improvements for segmentation on challenging MS-COCO with 5-shot setting in Table 2) method that decouples the standard classifier into two parallel heads (positive and negative heads) to independently process clear positive samples and noisy negative samples for mitigating the biased classification in FSOD and FSIS and thus improving the generalization ability of few-shot model on novel classes (and novel and base classes for generalized setting).
>
> Last but not least, our method **consistently outperforms its baseline and state of the arts by a large margin** recognized by you and other reviewers on few-shot benchmarks PASCAL VOC and MS-COCO for both FSOD/FSIS and gFSOD/gFSIS tasks without any additional computation cost and parameters.
>
> In summary, based on the above clarifications, we sincerely hope from the heart that you could re-evaluate the contribution of our work.
>
> **Q2: It is unclear why one of the person instances is mislabeled in Fig. 1.**
>
> Thank you for pointing out this issue. And we have some corresponding observations and discussions in Sec. 3.2: Missing label issue (lines 126-145). Here we provide more explanations. As we know, FSOD/FSIS tasks require instance-level recognition, which is different from image-level few-shot image classification. Note that the number of instances varies considerably because an image may consist of multiple instances.  In the standard benchmarks,  a few labeled instances are randomly sampled for given training images and shot numbers, and thus other unsampled instances (e.g., the person in Fig. 1) will be mislabeled.

---

> ### Author Response · Authors · 2022-08-07
> **further discussion**
>
> Dear reviewer LjLJ,
>
> Thank you for taking the time to review our work. We have provided corresponding responses, which we believe have covered your questions and concerns. We want to further discuss with you whether or not your concerns have been addressed.
>
> Best,

---

### Author Response · Authors · 2022-08-01
**Response to reviewers about FSOD/FSIS (gFSOD/gFSIS) benchmarks**

We thank all reviewers for reviewing our paper and their feedback, especially that they found the proposed method “**very simple yet effective**” (1weo, DeAx) and “**novel idea, quite interesting and strongly motivated**” (DeAx), “**solid framework**” (LjLJ), “**competitive performance** and **state-of-the-art results**” (LjLJ, 1weo, DeAx), “**adequate comparisons and evaluation**” (mATA), “**well analyzed and interpreted**” (DeAx) and “**clearly or well written**” (LjLJ, 1weo, DeAx). The main concern is the training and evaluation protocol (benchmarks) in few-shot detection and few-shot instance segmentation. We firstly clarify the training and evaluation protocol issues as follows：

First, we strictly follow the [standard training and evaluation protocol (benchmarks)](https://github.com/ucbdrive/few-shot-object-detection/blob/master/datasets/README.md) for FSOD/FSIS and gFSOD/gFSIS tasks which have been widely accepted and used in the community of machine learning and computer vision. To the best of our knowledge, recent published FSOD/FSIS (or gFSOD/gFSIS) papers almost utilize the same benchmarks that consider an object instance as a “shot”. The reason is that there are generally multiple instances in an image for instance-level FSOD and FSIS, which is different from image-level few-shot image classification.

Second, the community accepts the current benchmarks because they are more challenging for FSOD/FSIS (or gFSOD/gFSIS) tasks due to incomplete or partial annotations (i.e., missing labels) in a few-shot setting. The missing label issue requires that learning algorithms deal with training images each associated with multiple instances, among which only partial instances are labeled, which is also similar to partial label learning. As we know, missing label (partial label) learning is more difficult and challenging than conventional fully-supervised learning, especially few-shot scenarios.

Last but at least, we think that it is generally expensive and time-consuming to label all instances in many real-world applications. Fully-supervised object detection or instance segmentation typically assumes that all interest instances are fully labeled for given training images. In many real-world applications, such as open-vocabulary object detection, however, it is generally difficult to label all instances, and thus there still exists some instances left to be missing labeled. In addition, it may be more friendly and convenient for users to label partial instances than all instances even in few-shot scenarios, although we agree that it is possible to label all instances given few-shot images. The fully-annotated FSOD and FSIS are beyond our current research because we mainly focus on FSOD and FSIS in missing labeled few-shot scenarios. We leave fully-annotated FSOD and FSIS (creating new benchmarks) as future work.

We thank all reviewers for their time and efforts. Next, we respond to the concerns of each reviewer one by one.

---

### Meta-Review · Area_Chair_eWGA · 2022-08-29

**Recommendation:** Accept
**Confidence:** Less certain

**Metareview:**

**Summary**: This paper aims to address the missing label issue in few-shot object detection (FSOD) and instance segmentation (FSIS). In these tasks, some foreground examples (more specifically, classes) are not labeled in the training images, which causes classification bias for the conventional classification head. This paper proposes a simple yet effective algorithm that treats the foreground and background proposals differently in the classification loss. The proposed method is evaluated on benchmark datasets and shows competitive performance against SOTA methods for both FSOD and FSIS.

**Strength**: The paper is well-written. The observation is interesting and inspiring. The proposed method is novel, simple, effective, well-motivated, and compatible. The empirical study is solid and achieves SOTA results. The proposed method introduces no additional computation costs and hyper-parameters and is thus practical.

**Weakness**: The problem being solved is not well articulated. The inherent problem may come from poor task/protocol design. The paper might propose a solution to deal with the bias of existing benchmark datasets rather than the problem (FSOD/FSIS) themselves. Some of the paper's claims are not well-grounded. This missing label issue should be comprehensively analyzed from a broad perspective.

**Recommendation**: The paper received mixed review opinions from four reviewers. On the one hand, reviewers found the observation interesting and the proposed method simple, effective, and well-motivated. On the other hand, reviewers also have concerns that the missing label issue may not be the true problem of FSOD/FSIS, but a side effect of the poorly designed benchmark. Indeed, this was the main concern of the two reviewers (mATA and 1weo) who gave 3 and 4.

After the rebuttal, two reviewers are satisfied with the authors' responses to their concerns and raised their scores to 7. However, the above-mentioned two reviewers did not participate in the discussion.

The AC carefully read the paper and thought that the missing label issue can be a natural/practical issue in FSOD/FSIS. Specifically, when one wants to further detect an object class that is not in the base classes, one natural way to collect data is to find images that contain that object class and just annotate it (and ignore annotating other classes). Given this, the strengths of the paper, and the acceptance suggested by two reviewers, the AC suggests “acceptance.”

With that being said, the AC strongly suggests that the authors take the comments by reviewers mATA and 1weo seriously. Those questions, though might be beyond the scope, are quite valuable. For example, reviewer 1weo asks 1) if it is beneficial to collect more images (but each with partial labels) than fewer images (but each with full labels); 2) if other FSOD/FSIS methods work better or worse when only a few fully annotated images are provided. The AC respectfully thinks that the authors misunderstood the questions and unrelated responses.

Also, the authors' response to reviewer mATA can be improved. The authors said, “Second, the community accepts the current FSOD/FSIS benchmarks because they are more challenging due to incomplete or partial annotations (i.e., missing labels).” However, more *challenging* does not mean that the problem setting is appropriate/proper/practical. Similarly, *commonly used* benchmarks do not mean that they don't have problems in the setting. In response to reviewer mATA’s question, the authors could have focused more on why this setting is valid, practical, etc.

Overall, the AC sees the strength and value of the paper. To make the paper stronger and more impactful, the AC has the following suggestions. First, the AC suggests that the authors incorporate all the reviewers' comments and the authors' rebuttal into their final version. Second, the authors should add content that they promise to the reviewers. Third, the authors should add a paragraph to clarify the missing label issue. Fourth, the authors should further discuss the relationship to semi-supervised learning (e.g., the pseudo-label methods), which could potentially handle missing label issues.


**Award:**

No

---

### Decision · Program_Chairs · 2022-09-14

Accept